DOI: 10.1038/s41467-018-05945-4 | OPEN

# Loss of PRC1 induces higher-order opening of Hox loci independently of transcription during *Drosophila* embryogenesis

Thierry Cheutin[1] & Giacomo Cavalli [1]

Polycomb-group proteins are conserved chromatin factors that maintain the silencing of key developmental genes, notably the Hox gene clusters, outside of their expression domains. Depletion of Polycomb repressive complex 1 (PRC1) proteins typically results in chromatin unfolding, as well as ectopic transcription. To disentangle these two phenomena, here we analyze the temporal function of two PRC1 proteins, Polyhomeotic (Ph) and Polycomb (Pc), on Hox gene clusters during *Drosophila* embryogenesis. We show that the absence of Ph or Pc affects the higher-order chromatin folding of Hox clusters prior to ectopic Hox gene transcription, demonstrating that PRC1 primary function during early embryogenesis is to compact its target chromatin. Moreover, the differential effects of Ph and Pc on Hox cluster folding match the differences in ectopic Hox gene expression observed in these two mutants. Our data suggest that PRC1 maintains gene silencing by folding chromatin domains and impose architectural layer to gene regulation.

[1] Institute of Human Genetics, CNRS and the University of Montpellier, Montpellier, France. Correspondence and requests for materials should be addressed to T.C. (email: thierry.cheutin@igh.cnrs.fr) or to G.C. (email: giacomo.cavalli@igh.cnrs.fr)

PcG proteins are conserved epigenetic components, essential for cell differentiation, which maintain gene silencing during development[1–3]. Genome-wide studies have revealed large H3K27me3 chromatin domains bound by PcG proteins, and Polycomb (Pc) domains fold into distinct nuclear structures[4–8]. Hox gene clusters are the best characterized PcG targets and the loss of PcG proteins affects both Hox gene expression and chromatin compaction in *Drosophila* and mammalian cells[9–11]. In *Drosophila* embryos, Hox genes are grouped into two large chromatin clusters of 350–400 kb, covered by histone H3K27me3[4,12–14]. The Antennapedia complex (ANT-C) includes *lab*, *pb*, *Dfd*, *Scr*, and *Antp*, which control the cell identity of anterior segments, whereas the bithorax complex (BX-C) contains *Ubx*, *abdA*, and *AbdB* genes, which are responsible for the identity of posterior segments[15,16].

PcG proteins form two main classes of complexes, PRC2, which is responsible for the deposition of H3K27me3 of histone H3[17] and PRC1. PRC1 complexes are further subdivided into canonical PRC1 (cPRC1), which contains the Pc protein that binds to H3K27me3 via its chromo domain[18], and non-cPRC1 complexes, which lack Pc and contain other subunits[3]. In flies, Hox gene expression is regulated by cPRC1, which is composed of Sce, Psc-Suz2, and two proteins that are specific components of this complex: Ph and Pc. The mechanism by which cPRC1 mediates gene silencing is not understood. It has been shown that PRC1 is involved in chromatin compaction in vitro[19,20] and in vivo[10,11,21–25], but it is unknown whether PRC1-dependent transcriptional silencing is a consequence of its role on higher-order chromatin folding or whether higher-order chromatin compaction may instead be a consequence of silencing.

Here, in order to distinguish whether higher-order chromatin folding precedes PRC1-dependent transcriptional silencing, we analyze the time-course of 3D chromatin compaction and Hox gene expression in wild-type (WT) or in mutant embryos in which Ph or Pc are deleted. We show that the absence of Ph or Pc affects the higher-order chromatin folding of Hox clusters prior to ectopic Hox gene transcription, demonstrating that PRC1 primary function during early embryogenesis is to compact its target chromatin. During later embryogenesis, we observe further chromatin opening at Hox complexes in both Ph and Pc mutants, which is coupled to strong deregulation of Hox genes at this stage of development. Moreover, the differential effects of Ph and Pc on Hox cluster folding matches the differences in ectopic Hox gene expression observed in these two mutants, suggesting that the degree of Hox derepression in PcG mutants depends on the degree of structural constraints imposed by each PcG component.

## Results

**Correlation between Hox transcription and chromatin folding.** We first performed RNA fluorescence in situ hybridization (FISH) experiments in WT embryos to detect nascent transcripts of eight *Drosophila* Hox genes, using probes recognizing the first introns of those genes (Fig. 1a). As expected, our results show collinear expression of *lab*, *pb*, *Dfd*, *Scr*, *Antp*, *Ubx*, *abdA*, and *AbdB* along the anteroposterior axis[15,26] at the germ band elongated stage (3:50–7:20 after fertilization) (Fig. 1b, c; Supplementary Fig. 1), which is maintained throughout *Drosophila* embryogenesis (Supplementary Fig. 2). We then performed a series of immuno-DNA FISH experiments in embryos at the germ band elongated stage to address 3D chromatin compaction of Hox clusters. We measured 3D distances between FISH spots for *Ubx*, *abdA*, and *AbdB* of the BX-C cluster or *lab*, *Scr*, and *Antp* of the ANT-C cluster. The variation of these inter-spot distances along the anteroposterior axis shows that transcription of each Hox gene correlates with the opening of its corresponding

chromatin region, whereas the silenced portion of Hox complexes remains condensed (Fig. 1d–g). For example, when *Ubx* is activated in parasegment 5 (PS5) and PS6, the distances between the *Ubx* and *abdA* genes in these PSs were larger than those in the anterior PSs where all BX-C genes are silent. On the other hand, the distance *abdA*–*AbdB* did not increase in PS5-6 and is similar to that of more anterior PS (Fig. 1d), consistent with these two genes being silent in these regions of the embryo. In addition, Hox genes were found closer to Pc foci in PS where they were silenced than in their expression domains (Fig. 1h, i). Consistent with previous observations[9,27], these results indicate that repressed Hox genes are located in Pc foci when they are repressed and outside them when they are expressed. Although ChIP experiments have demonstrated binding of Ph and Pc at Hox genes in PS in which they were transcribed[28], our data show that the 3D folding of Hox complexes and Hox gene localization within Pc foci matches the Hox gene transcription patterns along the anteroposterior axis.

**Ph and Pc are required to maintain Hox genes silencing.** We then performed RNA FISH experiments in null mutant embryos for either the Ph (*ph^{del}*) or the Pc (*Pc^{XT109}*) subunits. In both mutants, Hox gene derepression started in a few cells, and the proportion of cells with derepression increased during later embryogenesis (Fig. 2; Supplementary Figs. 2–4). *Ubx* was the first gene of the BX-C cluster to be expressed ectopically in the anterior PS of *ph^{del}* embryos, whereas derepression of *abdA* and *AbdB* started later (Fig. 2a, f–h). Similarly, *Antp* was the first ANT-C gene to become derepressed in the head of both mutant embryos (Fig. 2i–k), whereas the others were derepressed at later stages (Supplementary Fig. 2). In both mutants, ectopic expression of each Hox gene depended on its position along the anteroposterior axis. For example, the number of cells showing ectopic *Ubx* expression decreased from PS4 to the anterior PS (Fig. 2a, b; Supplementary Fig. 2q and r), and *Scr* derepression was stronger in the posterior PS compared with that in the head (Fig. 2j). Taken together these results show that loss of Ph and Pc do not result in a general derepression of all Hox genes, although both proteins bind every Hox gene where they are repressed. In addition, ectopic Hox gene transcription was generally stronger and started earlier in *ph^{del}* embryos than in *Pc^{XT109}* embryos (Supplementary Fig. 2). For example, *Ubx* and *Antp* derepression occurred as early as 3:50–4:50 after fertilization in *ph^{del}* embryos, but only after the 4:50–6:00 stage of embryonic development in *Pc^{XT109}* (Fig. 2f, k). One exception was *AbdB*, which was derepressed earlier in *Pc^{XT109}* mutants than in *ph^{del}*, especially in PS7–PS12 (Fig. 2h; Supplementary Fig. 2w, x). This suggests that different cPRC1 subunits play specific roles on their target chromatin.

**Loss of Ph and Pc opens chromatin before transcription.** We then tested the possibility that cPRC1 mediates direct compaction of Hox clusters to prevent ectopic transcription. We reasoned that, if it does so, mutations in cPRC1 components would affect Hox compaction prior to any detectable transcriptional activation. To test this hypothesis, we performed DNA FISH experiments to monitor chromatin folding of the BX-C and ANT-C loci in cell nuclei of *ph^{del}* and *Pc^{XT109}* embryos (examples of DNA FISH with three signals from which we measured 3D distances, are shown in Supplementary Figs. 5–7). At the 3:50–4:50 stage after fertilization, distances between *Ubx*–*abdA*, *abdA*–*AbdB*, and *Ubx*–*abdA* were significantly increased in the head and PS0 of *ph^{del}* mutant embryos compared to those in control embryos (Fig. 3a–c), whereas neither *Ubx*, *abdA*, nor *AbdB* were derepressed (Fig. 2f–h). Similar general decompaction effects were

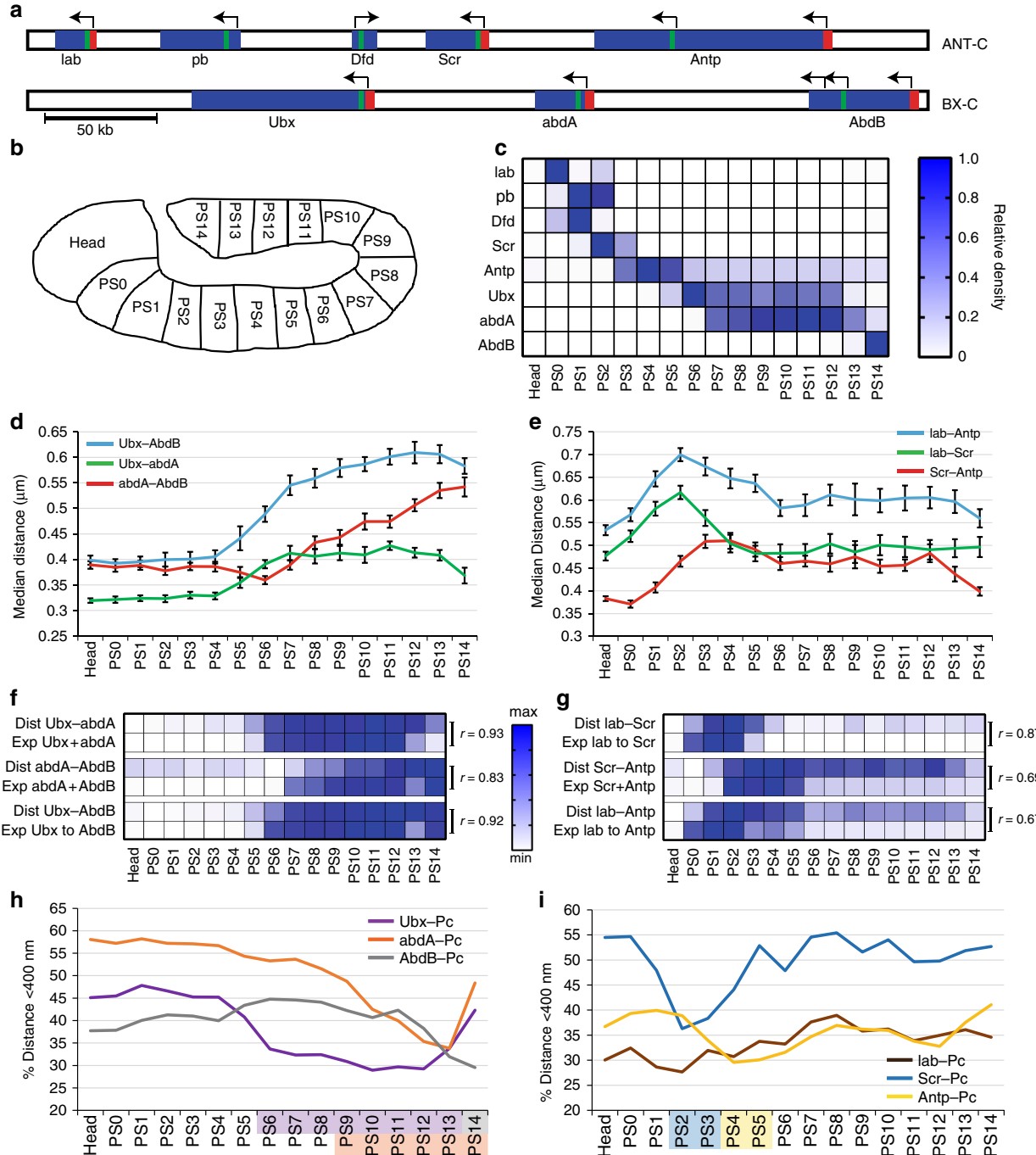

**Fig. 1** Hox gene expression correlates with Hox cluster chromatin opening. **a** Schematic representation of ANT-C and BX-C (Hox genes in blue, positions of DNA FISH probes in red, and positions of RNA FISH probes in green; the arrows show promoters). **b** Schematic diagram depicting the PSs of *Drosophila* embryos in which measurements were performed. **c** Relative density of RNA FISH spots measured for each Hox gene along the anteroposterior axis of WT *Drosophila* embryos 3:50–7:20 after fertilization. The density of RNA FISH spots is calculated by dividing the number of spots by the area of the PS. **d**, **e** Distances between the promoters of *Ubx*, *abdA*, and *AbdB* (d) or *lab*, *Scr*, and *Antp* (e) measured within cell nuclei of WT *Drosophila* embryos 3:50–7:20 after fertilization along the anteroposterior axis. We calculated distances between centroids of DNA FISH spots in three dimensions. For each PS of one embryo, we computed the median distances *Ubx–abdA*, *abdA–AbdB*, *Ubx–AbdB* or *lab–Scr*, *Scr–Antp*, *lab–Antp*. Curves represent the average median distances measured in several embryos and the corresponding error bar (SEM; N ≥ 16). **f**, **g** Heat maps showing correlations between Hox gene transcription and chromatin opening in the BX-C (**f**) and the ANT-C (**g**). The physical distance (Dist) between two loci within a Hox complex measured the chromatin opening and Hox gene expression (Exp) is calculated by adding the relative density of RNA FISH spot of Hox genes comprised between these two loci. All measurements were normalized between their minimum and their maximum. *r* indicates correlation coefficients. **h**, **i** Distances between promoters of *Ubx*, *abdA*, and *AbdB* (h) or *lab*, *Scr* and *Antp* (i) and the closest Polycomb foci were measured in WT *Drosophila* embryos 3:50–7:20 after fertilization. The percentage of distances measuring less than 400 nm was calculated for each PS of one embryo. Curves represent the mean percentage measured in several embryos. The highlighted PSs show regions wherein Hox genes were found to be significantly further away from Pc foci than in Head-PS0 (*t*-test, one-tailed, *P* < 0.01; *Ubx* in violet, *abdA* in orange, and *AbdB* in gray; *Scr* in blue and *Antp* in yellow)

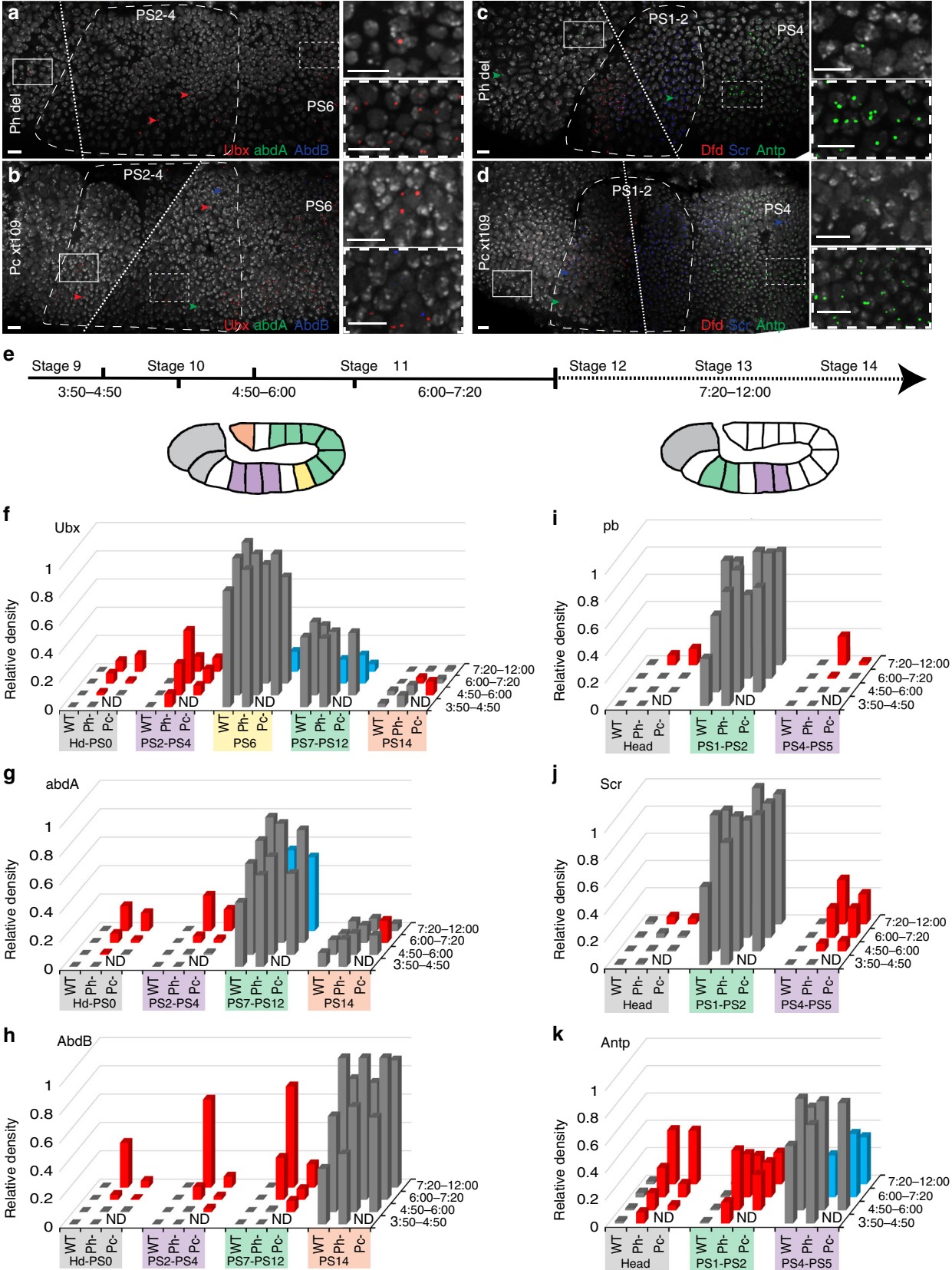

**Fig. 2** Timing of Hox gene derepression in $Ph^{del}$ and $Pc^{XT109}$ embryos. **a–d** RNA FISH images illustrating the earliest ectopic Hox gene expression observed in $ph^{del}$ (**a**, **c**) and $Pc^{XT109}$ (**b**, **d**) embryos. These images are maximum intensity projections of confocal images and dashed lines indicate where two images were merged. Arrowheads indicate few cells showing Hox gene transcription outside of their domains of expression (Ubx: red in **a**, **b**; abdA: green in **b**; AbdB: blue in **b**; Antp: green in **c**, **d** and Scr: blue in **d**). Scale bars, 10 μm. **e** Embryos were grouped into four classes based on their developmental stage, which depended on the duration of their development after fertilization at 25 °C. **f–k**, Relative densities of RNA FISH spots corresponding to Ubx (**f**), abdA (**g**), AbdB (**h**), pb (**i**), Scr (**j**), and Antp (**k**) expression measured in WT, $Ph^{del}$ and $Pc^{XT109}$ embryos during development. For simplicity, PSs wherein Hox genes of BX-C (**f–h**) and ANT-C (**i–k**) behave similarly were grouped (complete data are shown in Supplementary Figure 2). Red columns indicate where and when Hox gene transcription was significantly ectopically expressed in the mutants compared to WT embryos, whereas the blue columns show significant (Mann–Whitney U-test, two-tailed, $P < 0.01$) downregulation of Hox gene transcription

---

observed in PS2–PS4 (Fig. 3g–i), although abdA and AbdB are repressed and Ubx is derepressed only in a few cells of $ph^{del}$ embryos (Fig. 2f). Despite a weaker effect of Pc on BX-C folding, $Pc^{XT109}$ mutant embryos displayed significantly greater Ubx–abdA, abdA–AbdB, and Ubx–abdB distances than those of control embryos in PS2–PS4 at the 3:50–4:50 stage of development (Fig. 3g–i), without derepression of Ubx, abdA, and AbdB (Fig. 2f–h). Similarly, Ph and Pc were both required to globally compact ANT-C in the heads of embryos from the 3:50–4:50 stage after fertilization (Fig. 4a–c), whereas Antp was the only Hox gene of the ANT-C cluster to be derepressed, and this occurred only in $ph^{del}$ mutants (Fig. 2i–k). These results show that in PSs where every Hox gene of one complex is repressed, the first effect of Ph and Pc on Hox clusters folding can be detected before ectopic Hox genes transcription. Moreover, loss of Ph and Pc results in decondensation of the whole Hox clusters, whereas the first effects on Hox genes derepression affected a minority of the cells and onlys a few Hox genes.

To compare Ph and Pc, we plotted the effect of ph or Pc deletions on distances measured within the BX-C (Fig. 3d–f; j–l) or the ANT-C (Fig. 4d–f) during embryogenesis. After the 4:50–6:00 stage, the effect of both proteins on BX-C folding progressively increased (Fig. 3d–f; j–l) with a timing matching the ectopic expression of abdA and AbdB (Fig. 2g–h). At 7:20–12:00 h after fertilization, both mutant embryos showed a stronger opening of the BX-C in the head-PS0 (Fig. 3a–c) and PS2–PS4 (Fig. 3g–i). To summarize these effects, we plotted the three median distances between the promoters of Ubx, abdA, and AbdB (Fig. 3m–p) or between lab, Scr, and Antp (Fig. 4g–j). During early embryogenesis, the effects of the loss of Ph and Pc on Hox cluster folding were significant (Figs. 3a–c, n; 4a–c, h and Supplementary Fig. 8a–f). A stronger decompaction was observed in later embryogenesis (Fig. 3a–c, o; 4a–c, i and Supplementary Fig. 8g–l), consistent with strong ectopic Hox expression. These late effects coincide with the pattern of distance changes observed during physiological Hox activation in the appropriate PSs (Figs. 3p, 4j). Therefore, the strong effects on Hox distances observed in late development in the mutants is most likely due to the effect of ectopic transcription. Taken together, these results demonstrate that the effects of loss of PRC1 on condensation of Hox cluster chromatin precede transcriptional derepression. Therefore, chromatin opening in the mutants is not a consequence of transcription, suggesting that the primary function of PRC1 is to establish a compact architecture in cells where Hox loci are silenced.

**Ph and Pc only compact silenced Hox genes.** Since PRC1 was shown to bind to Hox genes also when they are active[28,29], we wondered whether the loss of PRC1 components would affect the higher-order organization of active Hox genes. To this aim, we investigated the consequences of Pc- or ph-null mutations on Hox loci. RNA FISH analysis did not reveal changes in Hox gene expression within their normal expression domains during early

embryogenesis (3:50–6:00 after fertilization) (Fig. 2f–k). We then analyzed chromatin compaction from DNA FISH data. In PS9–PS12, where Ubx and abdA are expressed but AbdB is silent in the WT, no significant effect on the Ubx–abdA distance was observed. However, as expected, the abdA–AbdB distance was increased in both $ph^{del}$ and $Pc^{XT109}$ embryos compared to control embryos (Fig. 5a, b). Conversely, the distance abdA–AbdB was not increased in PS14 where only AbdB is expressed, while the distance Ubx–abdA increased in both mutants during embryogenesis (Fig. 5c, d). Similarly, in the absence of Ph or Pc, the distance between the lab–Scr genes was significantly increased in PS4–PS5, where Hox genes located between lab and Scr are repressed in WT embryos (Fig. 5e, f). These results demonstrate that Pc and Ph compact chromatin fibers encompassing Hox genes only in cells in which they are normally repressed (Supplementary Figs. 9–10).

The fact that Pc was found to have a weaker effect than Ph on Hox clusters folding in the PSs where every Hox gene of each complex is repressed (Figs. 3 and 4) suggests that the two proteins might elicit different functions in cPRC1-dependent higher-order chromatin folding. We thus analyzed the nuclear distribution of each of the proteins in the presence of a null mutation in the other component. While the nuclear Pc distribution became diffuse in $ph^{del}$ embryos, Ph still accumulated in nuclear foci in $Pc^{XT109}$ embryos (Supplementary Fig. 11a–f). Furthermore, immuno-FISH experiments using anti-Ph and anti-Pc antibodies and FISH probes recognizing either abdA or Scr showed that Ph protein still accumulated at abdA and Scr loci in $Pc^{XT109}$ embryos, although its enrichment was weaker than in control embryos. This contrasts with the nuclear distribution of Pc, which did not accumulate on abdA and Scr in $ph^{del}$ embryos (Fig. 5g, h; Supplementary Fig. 11g, h). These data suggest that Ph retains ability to form higher-order structures in the absence of Pc to a greater extent than does Pc in the absence of Ph. Finally, we analyzed whether the degree of chromatin compaction induced by each of the proteins correlates with its effects on gene silencing. To this aim, for each PS where abdA or AbdB is repressed in WT embryos, we calculated the difference of abdA or AbdB expression between $ph^{del}$ and $Pc^{XT109}$ mutant embryos at stage 4:50–6:00. Scatterplots between these values and the difference of distance abdA–AbdB between $ph^{del}$ and $Pc^{XT109}$ mutant embryos at stage 3:50–4:50 showed a clear correlation between chromatin opening in early development and the subsequent ectopic transcription (Fig. 5i, j). These data suggest a causal link between chromatin condensation and gene silencing.

## Discussion

Our data indicate that deleting PRC1 components induces higher-order chromatin decompaction prior to ectopic transcription, suggesting that chromatin opening is not a consequence of transcription. This conclusion is valid in the assumption that RNA FISH is sensitive enough to detect the first events of ectopic gene expression. Several data suggest this to be the case. First, in

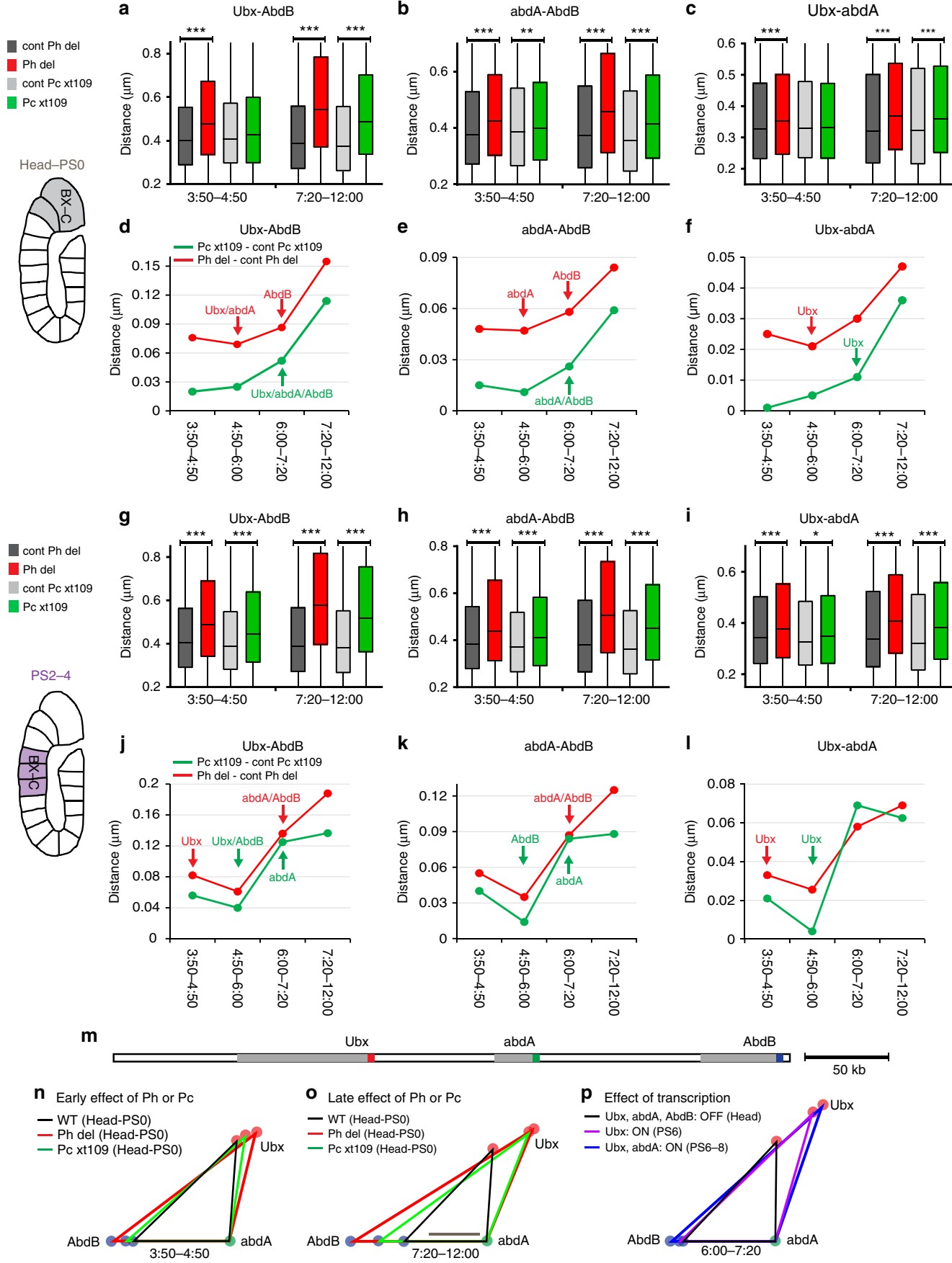

**Fig. 3** Ph and Pc are required to compact repressed BX-C before ectopic Hox gene transcription. **a–c**; **g–i** Box plots displaying distributions of the distances *Ubx–AbdB* (**a, g**), *abdA–AbdB* (**b, h**), *Ubx–abdA* (**c, i**), in the head–PS0 (**a–c**), and in PS2–PS4 (**g–i**) during early and late embryogenesis. Distances were measured in the cell nuclei of *ph*^del^ embryos (red) and their respective controls (dark gray) or *Pc*^XT109^ embryos (green), and their respective controls (light gray). Distance distributions are comprised between 0 and 1.5 μm and the lower and upper bounds of the colored rectangles correspond to the first and third quartiles, whereas the middle bars show the median distances. The black lines indicate significant differences between the mutants and their respective control embryos (Mann–Whitney *U*-test, two-tailed, \*$P < 0.05$; \*\*$P < 0.01$; \*\*\*$P < 0.001$). **d–f**; **j–l** Difference between the median distances *Ubx–AbdB* (**d, j**), *abdA–AbdB* (**e, k**), *Ubx–abdA* (**f–l**) measured in *ph*^del^ and the same distances measured in its control embryos (red) or in *Pc*^XT109^ and its control embryos (green), during embryonic development in head–PS0 (**d–f**) and PS2–PS4 (**j–l**). Arrows indicate when a Hox gene is firstly ectopically expressed in *ph*^del^ (red) or *Pc*^XT109^ (green) embryos. **m** Schematic linear representation of BC-X showing the DNA FISH probe positions. **n–p** Plots of the three median distances corrected for chromatic aberrations, between the promoters of *Ubx*, *abdA*, and *AbdB*. Comparison of the effects of Ph and Pc on the folding of BX-C during early (**n**) and late embryogenesis (**o**) with the openings induced by Hox gene transcription (**p**). Scale bar, 100 nm

order to detect early Hox genes transcription we used RNA FISH experiments with probes located in the first intron, i.e. we should be able to detect nascent transcripts within few minutes from the first productive event. Indeed, we observed RNA FISH signal in most of the cells inside embryonic areas where one Hox gene is expressed (Supplementary Fig. 1), indicating that this approach is sensitive and efficient. Furthermore, we identified ectopic Hox genes expression at earlier developmental stage compared to previous reports[30,31]. Similarly, we detected expression of dachshund in embryos at developmental stage 5 and vestigial at germ band extended stage (Supplementary Fig. 12a–b), in both cases earlier than previously reported[32,33] and corresponding to barely detectable transcription also using high-throughput approaches. Therefore, we believe that our RNA-FISH procedure readily detects low levels of gene expression. We also note that, while initial loss of silencing is detected in few cells of the corresponding PS, chromatin opening was generally observed in the whole embryonic region where each gene is normally silenced. Taken together, these results strongly indicate that cPRC1 compacts Hox clusters via the formation of higher-order chromosome structures during early *Drosophila* embryogenesis (Fig. 5k and Supplementary Movie 1). We thus propose that the absence of chromatin compaction in PcG mutant embryos does not directly trigger loss of silencing, but rather allows sequence-specific transcription factors to regulate Hox genes during later embryogenesis. The interplay between the general chromatin opening upon deletion of PRC1 and the identity and concentration of transcription factors that are able to bind to each of the Hox gene regulatory elements would ultimately determine the time and space of ectopic expression for each of the genes. For example, Hunchback has been shown to directly repress Ubx[34] and we observed a gradient of derepression of Ubx which is opposite to the Hunchback gradient. Therefore, we can speculate that the decrease of Hunchback expression during mid-embryogenesis, in conjunction with loss of PRC1, causes the observed pattern of ectopic *Ubx* expression in PcG mutant embryos.

We note that, during late embryogenesis when ectopic Hox gene expression occurs in some cells of one PS, an alternative approach to reveal chromatin opening in cells showing Hox gene derepression would be to combine DNA and RNA FISH experiments. However, applying this approach during early embryogenesis, which is the critical experiment in the present work, would not improve our results, since no cells show ectopic Hox gene expression at this developmental stage. Furthermore, in order to address Hox gene localization compared to Pc/Ph foci we used confocal microscopy. This approach allows us to study relative changes in distances between loci and Pc foci along the A/P axis, but its precision is limited. In future studies, it would be interesting to use super-resolution microscopy in order to measure the exact distance distributions. Despite these limitations, our results clearly indicate that the roles of the Ph and Pc proteins in the formation of PRC1 foci and Hox gene silencing are not

equivalent, with Ph showing stronger effects, except on the *abdA* and *AbdB* region of the BX-C. Since both Ph and Pc proteins were not detected in mutant Ph^del^ and Pc^xt109^ embryos from 3:50–4:50 stage after fertilization, it is unlikely that this difference results from maternal effects. In addition, a stronger maternal deposition of Pc or Ph could not explain why *Ubx* is derepressed later in *Pc*^XT109^ than in *ph*^del^ embryos, whereas the reverse is observed for *AbdB*. One possible explanation is that, in the absence of Pc and of its chromo domain, cPRC1 might only lose its anchoring to H3K27me3 while retaining some of its ability to bind discrete target regulatory elements and mediate their clustering through oligomerization of Ph[7,9,35] (Supplementary Fig. 12c–d)[36]. The effects of *Pc* deletion would thus depend on the levels of H3K27me3 at each locus and, indeed, H3K27me3 levels are highest in the *abdA–AbdB* region of the BX-C compared to all others[28]. On the other hand, in the absence of Ph, cPRC1 is expected to lose its ability to form higher-order structures through oligomerization, therefore inducing strong decompaction throughout the BX-C and ANT-C and, consequently, a strong loss of silencing. Further studies will be required to elucidate this point and the mechanism of PRC1-mediated silencing at other genes. In particular, it will be interesting to test whether the role of PRC1 in chromatin condensation is predominant at large Pc domains containing many PRC1-binding sites and whether the mechanisms of silencing differ at smaller target loci, both in *Drosophila* and in mammals.

## Methods

**Fly lines and embryo fixation.** The Oregon-R w[1118] line was used as the WT control line. The *ph*^del^ stock is a null mutant[37] and was balanced over the *KrGFP-FM7c* balancer (FKG: obtained from BL#5193 of the Bloomington *Drosophila* Stock Center). The *Pc*^XT109^ stock is a null mutant[38] and was balanced over the *KrGFP-TM3, Sb* balancer (TKG: obtained from BL#5195 of the Bloomington *Drosophila* Stock Center). Flies were maintained on standard cornmeal yeast extract media at 21 °C. Embryos were harvested on agar/vinegar plates. Embryos were fixed according to the protocol described by Bantignies and Cavalli[39]. Briefly, embryos were dechorionated with bleach for 5 min and transferred into a glass flask, followed by addition of 5 ml of fixation buffer containing: 4% paraformaldehyde, KCl (60 mM), NaCl (15 mM), spermidine (0.5 mM), spermine (0.15 mM), EDTA (2 mM), EGTA (0.5 mM), PIPES (15 mM), and 5 ml of heptane. Embryos were fixed under vigorous agitation on mini-shaker for 25 min. The aqueous phase was then removed and 5 ml of methanol was added. After shaking for 1 min, fixed embryos were collected at the bottom of the glass flask and stored in methanol at −20 °C.

**RNA FISH.** RNA FISH experiments were performed according to the protocol described by Kosman et al.[26]. Probes were prepared using a direct labeling approach with a FISH Tag RNA Kit (Thermo Fisher Scientific, F32956), and primers used to synthesize RNA probes are listed in Supplementary Table 1. We used three fluorochromes: A488 (~14 pmol/μl for a RNA concentration of ~75 ng/μl), A555 (~3.4 pmol/μl for a RNA concentration of ~50 ng/μl), and A647 (~2.3 pmol/μl for a RNA concentration of ~35 ng/μl). Briefly, embryos were transferred from methanol to ethanol and then incubated for 1 h in 90% xylene/10% ethanol. After being washed sequentially in ethanol and methanol, embryos were incubated for 25 min in PBS + 0.1% tween (PBT) + 5% formaldehyde. After being washed in PBT, the embryos were treated with a solution of proteinase K (10 μg/ml) in PBT for 5–6 min. This reaction was terminated by washing the embryos in PBT. A

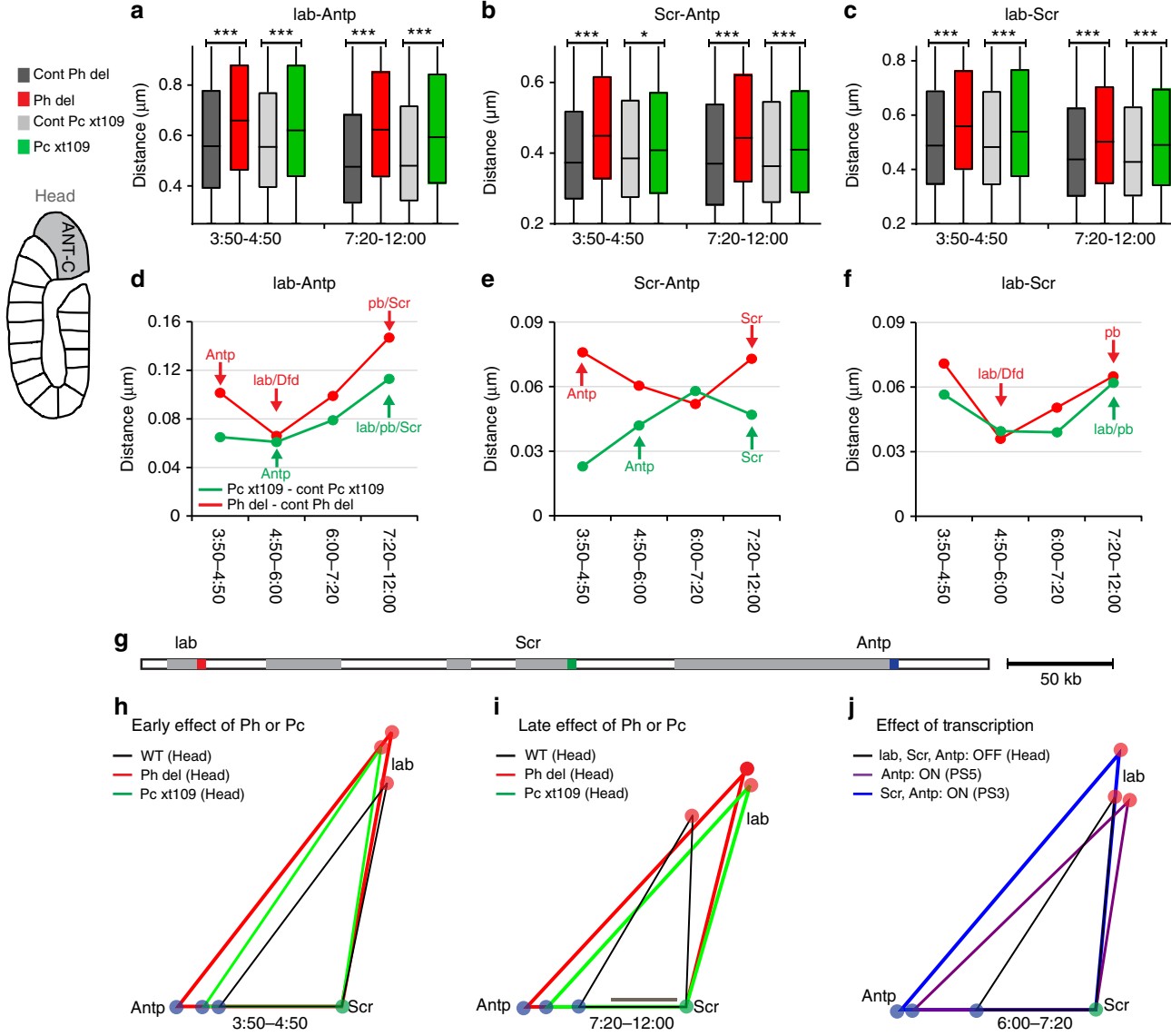

**Fig. 4** Ph and Pc are required to compact repressed ANT-C before ectopic Hox gene transcription. **a–c** Box plots displaying distributions of the distances: *lab–Antp* (**a**), *Scr–Antp* (**b**) and *lab–Scr* (**c**) in the head during early and late embryogenesis. Distances were measured in the cell nuclei of *ph^del* embryos (red) and their respective controls (dark gray) or *Pc^XT109* embryos (green) and their respective controls (light gray). Distance distributions are comprised between 0 and 1.5 μm and the lower and upper bounds of the colored rectangles correspond to the first and third quartiles, whereas the middle bars show the median distances. The black lines indicate significant differences between the mutants and their respective control embryos (Mann–Whitney *U*-test, two-tailed, *P < 0.05; **P < 0.01; ***P < 0.001). **d–f** Difference between the median distances *lab–Antp* (**d**), *Scr–Antp* (**e**), and *lab–Scr* (**f**) measured in *ph^del* and the same distances measured in its control embryos (red) or in *Pc^XT109* and its control embryos (green) during embryonic development in head. Arrows indicate when a Hox gene is firstly ectopically expressed in *ph^del* (red) or *Pc^XT109* (green) embryos. **g** Schematic linear representation of ANT-C showing the DNA FISH probe positions. **h–j** Plots of the three median distances corrected for chromatic aberrations, between the promoters of *lab*, *Scr*, and *Antp*. Comparison of the effects of Ph and Pc on the folding of ANT-C inside cell nuclei during early (**h**) and late embryogenesis (**i**) with the openings induced by Hox gene transcription (**j**). Scale bar, 100 nm

second post-fixation step was then performed with PBT + 5% formaldehyde for 25 min. After being washed in PBT, embryos were transferred to a hybridization solution (50% formamide, 5 × SSC, 100 μg/ml tRNA, 50 μg/ml heparin, and 0.1% Tween), and pre-hybridization was performed for 1 h at 55 °C. Finally, embryos were hybridized for 16–20 h at 55 °C with labeled probes followed by extensive washes. Typically, we added 1.2 μl of three probes containing either A488, A555, or A647 to 50 μl of hybridization buffer. DAPI staining was performed before mounting the embryos in Vectashield (Eurobio, H-1000) between a slide and coverslip.

RNA FISH experiments were performed on embryos collected from Oregon-R w^1118, *ph^del*/FKG, or *Pc^XT109*/TKG flies. Ectopic Hox gene expression was never observed in Oregon-R w^1118 embryos, whereas ~25% of the embryos issued from Ph^del/FKG crosses showed ectopic expression of *Ubx* or *Antp* from developmental stage 9 through later embryogenesis and corresponded to *ph^del* mutant embryos. Similarly, ~25% of the embryos issued from *Pc^XT109*/TKG crosses showed ectopic

expression of *Ubx*, *AbdB*, or *Antp* at developmental stage 10 through later embryogenesis and corresponded to *Pc^XT109* mutant embryos. Both balancer chromosomes FKG and TKG did not affect Hox genes expression since no difference have been observed between WT, Ph^del control, and Pc^xt109 control embryos.

**Immuno-DNA FISH and immuno-localization**. Immuno-DNA FISH experiments were performed according to the protocol described by Bantignies and Cavalli[39]. Briefly, to re-hydrate fixed embryos, we used a series of methanol/PBS + 0.1% tween (PBT) solutions: 100/0; 90/10; 70/30; 50/50; 30/70; 0/100. Embryos were then incubated in a solution of RNAseA (200 μg/ml) in PBT for 2 h. After an additional 2 h incubation in PBS + 0.2% triton X-100 (PBTr), embryos were progressively transferred in pre-hybridization mixture (pHM) containing 50% formamide, 4 × SSC, 100 mM NaH₂PO₄, and 0.1% tween. Embryos were incubated

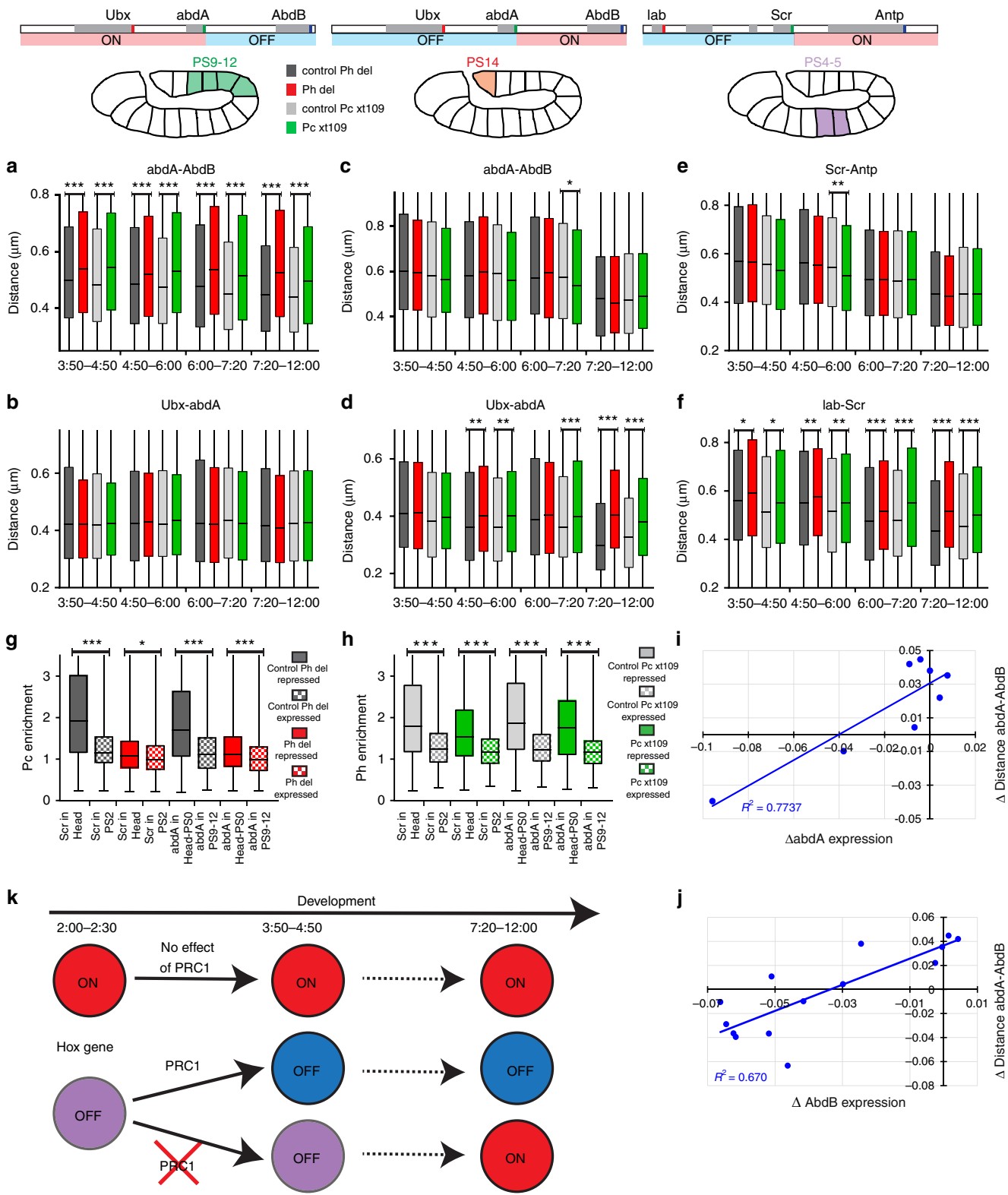

20 min in a series of PBTr/pHM solutions: 80/20; 50/50; 20/80; 0/100. A denaturation step was done by incubating embryos in pHM solution for 15 min at 80 °C. Then, hybridization was performed for 16–20 h at 37 °C in a solution containing deionized 50% formamide, 2 × SSC, 10% dextransulfat, 0.5 mg/ml salmon sperm DNA, and labeled probes. Embryos were washed twice for 20 min at 37 °C in a solution containing 50% formamide, 2 × SSC, and 0.3% CHAPS. Finally, embryos were transferred in PBT solution by progressively decreasing the

concentration of formamide. Primers used to synthesize DNA FISH probes hybridizing with the promoters of *Ubx*, *abdA*, *AbdB*, *lab*, *Scr*, and *Antp* are listed in Supplementary Table 1. Immuno-DNA FISH experiments were performed using either an anti-Pc rabbit polyclonal antibody (dilution 1/200) or an anti-Ph goat polyclonal antibody (dilution 1/500) developed in our laboratory[40] as primary detection antibodies with A405-conjugated secondary antibodies (anti-rabbit: Thermo Fisher Scientific; A-31556; dilution 1/200 or anti-goat: abcam; Ab 175664;

**Fig. 5** cPRC1 only compacts chromatin in the repressed regions of the BX-C and ANT-C domains. **a–f** Box plots displaying distributions of the distances *abdA–AbdB* (**a**, **c**), *Ubx–abdA* (**b**, **d**), *Scr–Antp* (**e**), and *lab–Scr* (**f**) measured in the cell nuclei of *Ph^del* embryos (red) and their respective controls (dark gray) or *Pc^XT109* embryos (green) and their respective controls (light gray). Measurements were made in PS9–12 (**a**, **b**), PS14 (**c**, **d**), and PS4–5 (**e**, **f**) during development. Distances distribution are comprised between 0 and 1.5 μm. **g**, **h** Box plots presenting Pc (**g**) and Ph (**h**) enrichments at *Scr* and *abdA* FISH spots. Distribution of Pc and Ph enrichments are comprised between 0.2 and 11.33. The lower and upper bounds of the colored rectangles correspond to the first and third quartiles, whereas the middle bars indicate the median distances (**a–h**). Significant differences are indicated (Mann–Whitney, *U*-test, two-tailed, *$P < 0.05$; **$P < 0.01$; ***$P < 0.001$) (**a–h**). **i**, **j** Scatterplots showing correlations between the differential effect of Ph and Pc on the distance *abdA–AbdB* at 3:50–4:50 and the differential effect of Ph and Pc on *abdA* or *AbdB* expression at 4:50–6:00. Each point corresponds to one PS where *abdA* (**i**) or *AbdB* (**j**) are repressed in WT embryos. **k** Schematic diagram summarizing the effects of PRC1 on Hox gene folding and transcription. Circles represent silenced (OFF) or transcribed (ON) chromatin associated with Hox genes (red, open; purple, partly compact; blue fully compact). cPRC1 has no effect when Hox genes are expressed and chromatin is open. When Hox genes are repressed, cPRC1 compacts their chromatin during early embryogenesis and they will remain silenced. Without cPRC1, this compaction cannot occur and silenced Hox genes might become subsequently transcribed

dilution 1/200). FISH probes were directly labeled with A555 (~6.4 pmol/μl for a DNA concentration of ~70 ng/μl, *Ubx* or *lab*), A488 (~11 pmol/μl for a DNA concentration of ~75 ng/μl, *abdA* or *Scr*), or A647 (~3.6 pmol/μl for a DNA concentration of ~50 ng/μl, *AbdB* or *Antp*). Typically, we added 1.2 μl of three probes containing either A488, A555, or A647 to 50 μl of hybridization buffer. Immuno-FISH experiments were performed on embryos collected from Oregon-R w^1118, *ph^del*/FKG, or *Pc^XT109*/TKG flies. Ph and Pc immunolabelling, respectively, allowed us to discriminate mutants from controls since ~25% of the embryos were devoid of Ph or Pc from developmental stage 9 to the end of embryogenesis. Therefore, distance measurements done in embryos lacking Ph or Pc correspond to results of mutant embryos, whereas distance measurements done in embryos with normal levels of Ph or Pc were used as controls. Both balancer chromosomes FKG and TKG did not affect the folding of Hox gene clusters since no difference have been observed between WT, Ph^del control, and Pc^xt109 control embryos. To evaluate chromatic aberrations, we performed FISH experiments with three probes (A448, A555, and A647) that hybridized to only one locus (*Antp*), and we measured the distances between the three different labels. The median distance between A488 and A555 was 130 nm, between A555 and A647 was 120 nm, and between A488 and A647 was 195 nm.

We used A488-conjugated anti-rabbit (Thermo Fisher Scientific; A-21206; dilution 1/200) and A555-conjugated anti-goat (Thermo Fisher Scientific; A-21432; dilution 1/200) secondary antibodies for Pc and Ph double immunolabelling experiments. Pc/Ph immuno-DNA FISH experiments were performed using either *Scr* or *abdA* DNA probes coupled to A488. We used A555-conjugated anti-rabbit (Thermo Fisher Scientific; A-31572; dilution 1/200) and A647-conjugated anti-goat (Thermo Fisher Scientific; A-21447; dilution 1/200) secondary antibodies to detect anti-Pc and anti-Ph primary antibodies.

**Microscopy and image analysis**. Control and mutant genotypes were imaged and analyzed by using the same parameters in all the experiments performed in this work. Images were collected using a LSM 780 microscope (Carl Zeiss Microscopy, Iena) with a GaAsP detector and a ×60 numerical aperture (NA) 1.4 objective. Images of RNA FISH and double Ph/Pc immunolabelling were acquired with a pixel size of 110 nm and a z-step of 0.5 μm. I-FISH experimental images had pixels of 73 nm and a z-step of 0.3 μm to measure the 3D distance between three FISH signals, whereas double Pc/Ph immunolabelling coupled with single FISH (Fig. 5g, h and Supplementary Fig. 11g, h) were acquired with a pixel size of 94 nm and a z-step of 0.3 μm. RNA FISH experiments were quantified using ImageJ software. First, a Gaussian filter with a radius of 1 was applied to reduce noise. Z-projection was then calculated using maximum intensity projection. Second, areas corresponding to the PSs of interest were manually drawn. We then measured their surface area and counted the number of FISH spots using the option "find maxima".

Distances between DNA FISH signals were measured using Volocity software (Perkin Elmer, Coventry). A fine (3 × 3 × 3 pixels) filter was applied to reduce noise, and the three FISH signals were segmented using the threshold option "get objects". PSs were drawn manually, and we measured distances between the centers of mass of segmented objects in each PS of one embryo. For each object corresponding to one FISH spot, we computed the two distances with the two closest objects of the two other FISH signals and we only kept measurements when these two distances were below 1.5 μm. Therefore, measurements have only been made when one FISH spot for the three different FISH probes can be detected within a radius below 1.5 μm. In this study, distance measurements have been presented in two ways. On the one hand, distances coming from several embryos of the same conditions were pooled (i.e. same PS, same developmental stage and same type of embryos). We then plotted their global distributions comprised between 0 and 1.5 μm and used Mann–Whitney *U*-test to compare their distributions obtained in two different conditions (Figs. 3 and 5; Supplementary Fig. 8). On the other hand, we computed the median distances *Ubx–AbdB*, *abdA–AbdB*, and *Ubx–abdA* (Fig. 1; Supplementary Fig. 9) or *lab–Scr*, *Scr–Antp*, and *lab–Antp* (Fig. 1; Supplementary Fig. 10) for each PS of one embryo. Median distances have been calculated in several embryos of the same condition, and we then plotted their

mean along the A/P axis. Here, we used the *t*-test to compare two conditions. Both ways of analyzing distance measurements give the same conclusions. To globally visualize the effects of Ph, Pc, and Hox gene transcription on folding of BX-C and ANT-C, median distances were corrected for chromatic aberrations (Fig. 3n–p and Fig. 4h–j). Images shown in Supplementary Fig. 5–7 were done with the option "3D opacity" and the mode "maximum intensity" of the Volocity software.

To quantify the enrichment of Pc/Ph in nuclear foci (Supplementary Fig. 11), we used Image-J software to perform a Gaussian filter (radius = 1) and a 3 μm-thick maximum intensity projection. Then, we computed local maxima and average nuclear intensity in 25 nuclei located in the head. Similarly, we used Image-J software to quantify Pc/Ph enrichment at *Scr* and *abdA* loci (Fig. 5g, h). We performed a Gaussian filter (radius = 1) and used the FISH channel to compute local maximum corresponding to FISH spots. Then, we measured the pixel intensity in the channel corresponding to Ph or Pc immunolabelling at the local maxima previously identified. We divided the later intensity with the average nuclear intensity to calculate Ph or Pc enrichments. Finally, we used Mann–Whitney *U*-test to compare distributions obtained in two different conditions.

## Data availability

The data sets generated during the current study are available from the corresponding authors upon request.

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

## Acknowledgements

We acknowledge the *Drosophila* and imaging MRI facilities, member of the National Infrastructure France-BioImaging supported by the French National Research Agency (ANR-10-INBS-04, "Investments for the future"). We thank Satish Sati and Frédéric Bantignies for critical reading of the manuscript. G.C. Research in the laboratory of G.C. was supported by grants from the European Research Council (ERC-2008-AdG no. 232947), the CNRS, the FP7 European Network of Excellence EpiGeneSys, the European Union's Horizon 2020 research and innovation program under grant agreement No. 676556 (MuG), the Agence Nationale de la Recherche (N. ANR-15-CE12-006-01), the Fondation pour la Recherche Médicale (Grant no. DEI20151234396), the French National Cancer Institute (INCa, Project no. PLBIO16-222), and the Laboratory of Excellence EpiGenMed (No. ANR-10-LABX-12).

## Author contributions

T.C. and G.C. initiated and led the project. T.C. performed experiments and data analysis. T.C. and G.C. interpreted and discussed the data and wrote the manuscript.Data availabilityThe data sets generated during the current study are available from the corresponding authors upon request.

## Additional information

**Competing interests:** The authors declare no competing interests.

