## [Peer Review File · Nature Communications]

Reviewers' comments:

Reviewer #1 (Remarks to the Author):

Genes silenced by Polycomb group (PcG) complexes are known to be more compact than actively transcribed genes. The PcG complex PRC1 drives this compaction. Is this compaction correlated with gene silencing? The authors present a series of experiments that show that, when PRC1 components are removed, chromatin becomes decompact prior to gene expression. The results show that loss of compaction itself does not necessarily lead to gene expression, but it does precede it, showing there is a correlation between compaction and gene silencing. The question addressed in this manuscript is an important one and the experimental approaches they used are appropriate. Below are a number of questions and suggestions to improve the overall quality of the manuscript.

1. Fig. 1a-d - The FISH spots are almost invisible in the manuscript provided (even the source file). The poor quality of these images drives the reader to think "how were distances between the DNA-FISH spots measured?" The authors need to show zoomed insets of a few cells where two DNA-FISH spots are visible with a line indicating how the spot distances were measured. Also, it is difficult to determine where the parasegments are located. It is mentioned in the materials and methods that regions corresponding to the parasegments of interest were drawn manually. It would greatly help the reader to see these regions of interest outlined in a white dotted line.
2. Fig. 2 and 3- The labels "Pc xt109-control" and "Ph del-control" are confusing. Are the authors referring to the control genotype or the mutant minus the control genotype? It would be less confusing and more informative to show both mutant and control on the graphs. In the figure legends where the triangle plots are described as showing "the folding of BX-C" is confusing. A better wording would be "distance between BX-C genes."
3. Fig. 4g - The genotypes and thus the colors, are in a different order than the previous six graphs. This confuses the reader, be consistent.
4. Line 10 and line 41 - references are needed for the statement "ectopic transcription can open chromatin"
5. Line 71-75- authors show that ectopic Hox gene expression generally started earlier in Phdel embryos than in Pcxt109 embryos, but there is not any expression analysis of the Hox genes in Pcxt109 embryos for the early timepoint 3:50-4:50 hours after fertilization (Fig. 1 and Fig. Extended Data).
6. Line 127- authors claim that the nuclear Pc distribution became diffused in Phdel embryos whereas Ph still accumulates in foci in Pcxt109. This seems to be an overstatement from the image they provided (extended data 7 a-c). The number of Ph foci is drastically reduced from WT to Pcxt109. The authors need to show quantification of the Ph and Pc spots in WT vs mutants in several embryos.
7. Fig. S1 legend - Mention that a) the images shown are maximum projections of confocal images, b) FISH spots are measured in three dimensions and c) that the density of FISH spots is measured by area of the parasegment.
8. Fig. S3 - Conclusions cannot be drawn from the poor quality images provided.
9. Supp. Methods RNA FISH - List the fluorophores used for labeling probes. Please provide the final concentration of probe used for the hybridization in pmol fluor/ μ l.
10. Supp. Methods Immuno-DNA FISH - What are the dilutions used for each antibody? What company produced the secondary antibodies used in the experiments?
11. Supp. Methods microscopy and image analysis - Were the control and mutant genotypes were imaged and analyzed under the same parameters? This is critical to the integrity of the conclusions regarding FISH spot density, as different densities could be obtained by using different imaging parameters for control and mutant samples.

Reviewer #2 (Remarks to the Author):

In this study authors presented evidence describing parasegment-specific transcriptional and chromatin structure at HOX clusters during *Drosophila melanogaster* embryogenesis. Using RNA-FISH and DNA-FISH authors followed in time and space the activation of HOX genes and the relative higher order chromatin conformation. Further, they performed the same analysis on two PRC1 mutants, describing alterations both at the transcriptional and structural levels. From the data analysis authors concluded that PcG dependent chromatin structure dynamics are not associated to observed transcriptional changes.

This work is interesting, potentially unravelling the link between PRC1-mediated chromatin architecture and transcriptional activation. I have some concerns about the extent to which the current molecular data support the final conclusion. In fact, the main conclusion of this work depends on a comparison between RNA-FISH and DNA-FISH, experimentally performed in parallel. While it is reasonable to compare the same RNA-FISH analysis on different strains, the conclusions about chromatin structure and transcription can not be drawn only on the basis of DNA-FISH/RNA-FISH comparison because the two technologies could have different resolutions. Moreover, DNA-FISH data was presented as an average of the minimal distances between two spots in the parasegment-specific population and did not provide an estimation at the single cell level of the percentage of nuclei lacking DNA interaction (images were not presented!). A combo DNA/RNA FISH could prove that in the PcG mutants, nuclei presenting different chromatin conformations do not show local transcription. Considering that the work did not provide strong evidence for its conclusions I cannot support a publication in Nature Communication.

Other criticisms:

1. In RNA-FISH, absence of the spot does not prove the absence of transcription, thus authors could underestimate a subpopulation of cells with lower transcription in PcG mutants. Parasegment microdissections followed by single cell RNA analysis could untangle this point.
2. I cannot find in the figure legend or in the methods the number of nuclei that were taken into consideration in the RNA-FISH and DNA-FISH analyses. Only the number of embryos is indicated.
3. Extended Fig 3 h and j: I am not sure that the green signal is, as expected, inside nuclei.
4. Extended Fig 3: the image "j" is cut.
5. Images showing DNA-FISH were showed only in extended data Figure 7 with only one probe. Why? The authors should document their experiments and quantification with representative images.
6. Figure 4g: Authors decided to normalize the Pc signal with "the average intensity inside the cell nuclei". However, they described a diffuse Pc signal in Ph mutant in the extended data Figure 7b. Thus the normalization inside nucleus could have a bias.
7. The Pc diffusion showed in the extended data Figure 7b is not present in the same strain in extended data Figure 7d.
8. Super resolution analysis could improve the quality of immuno-FISH experiments.
9. As control authors should measure the localization of Pc in FISH analysis of HOX regions that does not change conformation, such as AbdA-AbdB.
10. An alternative experiment that can measure the amount of Pc protein in the close proximity of DNA of interest is a variant of the Proximity Ligation Assay (PLA) (Gustafsdottir, SM et al., PNAS 2007), used to quantify Protein/DNA binding in immunofluorescence.

Reviewer #3 (Remarks to the Author):

Summary:

In this study, the authors undertook a massive exercise to image Hox cluster loci in *Drosophila*

through early stages of embryogenesis. They compared the effect of deleting two different components of canonical PRC1 – Ph and Pc – on intergene-distance between Hox cluster genes. The main outcome is that the authors were nicely able to separate two known functions of PRC1 – chromatin compaction and gene repression, in time. They show that decompaction of Hox cluster genes precedes transcriptional activation. Further they confirmed the conclusion previously made by them and by other groups that PRC1 loss leads to decompaction of the large Hox gene clusters. This is an important addition to the literature concerning the mechanisms that repress expression to control developmental progression.

Comments:

1. This is an elaborately designed study with several time points and a lot of data. The data, however, are presented in a manner that is hard to absorb. Thus, it takes the reader a significant amount of effort to arrive at the conclusions that the authors make. One issue is that the expression data (most of Fig. 1) is presented in three dimensional histograms, while the compaction data (Figs. 2 and 3) is presented in box plots and graphs. The disparate means of presentation makes direct comparison hard to do. Perhaps the authors might take especially central examples and do a plot of expression vs compaction side by side in the same format.
2. The actual microscopy images in Figure 1 are nearly impossible to appreciate and one has to rely entirely on the quantification. The entire study relies heavily on measurements of intergene distance between Hox genes. It is important to show the reader examples of those measurements, and what the primary data looks like when comparing a compacted and a more open setting.
3. The title suggests that compaction has a causal role in preventing ectopic transcription. While the authors have convincingly demonstrated that Hox decompaction precedes Hox transcription, I do not think that we have learned from this study that compaction causes gene silencing or prevents gene expression. As the author's themselves find and later suggest, the decompacted Hox genes are not immediately expressed, probably due to the lack of an appropriate activator. Perhaps expand on this topic and offer alternative explanations.
4. This is not necessary for this study, and technically a very tough thing to do, but it would be nice and very interesting to see if the general principles described in this study are unique to the uniquely organized Hox genes or whether they are also true for other, more typically organized, PRC1 target loci.

Reviewer #4 (Remarks to the Author):

"PRC1-dependent compaction of Hox gene clusters prevents transcriptional derepression during early *Drosophila* embryogenesis" by Thierry Cheutin and Giacomo Cavalli is an impressively detailed study that is even more remarkable given the small number of authors (two). It is focused on a very specific question, which is whether the repressive effect of PRC1 on gene activity is the result of chromatin compaction. To address this question, the authors examine the temporal relationship between chromatin compaction and gene silencing/activation at the bithorax and Antennapedia complexes (BX-C and ANT-C) in wild-type embryos as well as in embryos mutant for either polyhomeotic (ph) or Polycomb (Pc). The authors report a correlation between chromatin compaction and gene silencing and, furthermore, that the null state of ph or Pc leads to chromatin decompaction prior to the activation of Hox gene expression from the BX-C and ANT-C. These observations lead the authors to conclude that the mechanism by which PRC1 represses gene expression is through the compaction of chromatin. The figures show meticulous data from countless DNA and RNA fluorescent in situ hybridization (FISH) assays targeting regions within the BX-C and ANT-C and, overall, the narrative is well-written. Even so, there are a number of issues that deserve attention from the authors. These are noted below:

- 1) The most pressing technical issue is whether the method used to assess transcriptional activity is sensitive enough to justify the strength of the conclusions. Thus, the authors are encouraged to quantify, and/or consider in detail, the sensitivity of their assays and then adjust the strength of their conclusions accordingly. For example, how certain are the authors that their assays could

have detected a very low level of transcription or perhaps even paused polymerases?

2) Similarly, the underlying logic of the study - that decompaction prior to gene activation in a mutant background demonstrates that chromatin compaction is the mechanism for silencing - is not as strong as the authors argue it to be. For example, formally speaking, PRC1 might independently induce to two outcomes, one being chromatin compaction and the other being gene silencing. In this scenario, disruption of PRC1 would be expected to result in both decompaction and transcriptional activation, but without a necessary relative temporal relationship. Thus, the authors are urged to provide a more extensive background (including primary references) to the decades-old question of how Hox genes are regulated, a more nuanced explanation of their logic, and a more balanced interpretation of the data. The following are examples of statements in need of better argument or balance:

a) "In summary, our data demonstrate that binding of PRC1 to large genomic domains during early embryogenesis induces the formation of compact chromatin to prevent ectopic gene expression at later time-points. Thus, epigenetic mechanisms such as Polycomb mediated silencing act by folding chromatin domains and impose an architectural layer to gene regulation."

b) "Therefore, the strong effects on Hox distances observed in late development in the mutants is most likely due to the effect of ectopic transcription. Taken together, these results demonstrate that the loss of PRC1 prevents the condensation of Hox clusters prior to any transcriptional derepression. Thus, chromatin opening in the mutants is not a consequence of transcription, suggesting that the primary function of PRC1 is to establish a compact architecture in cells where Hox loci are silenced."

c) "These results demonstrate that Pc and Ph compact chromatin fibres encompassing Hox genes only in cells in which they are repressed (Extended Data Fig. 5-6)."

d) "Taken together, these results demonstrate that cPRC1 compacts Hox clusters via the formation of higher-order chromosome structures during early Drosophila embryogenesis (Fig. 4j)."

3) There are two instances where expression in ph(del) embryos seems to have been detected at the earliest time point (3:50 - 4:50 hr AEL): Ubx in PS2-4 and Antp in the head. As these observations are so directly relevant to the conclusions, have the authors looked at an earlier time point? If not, the authors may wish to consider a more serious discussion of these exceptions.

4) Will the authors please comment on the potential relevance of maternal effects to the interpretation of their data and their conclusions? In addition, might maternal effects explain the more obvious effects of the mutant backgrounds later in embryogenesis?

5) Please provide a better description of ph(del) and PC(XT109), as these are critically important mutations. In particular, it would be helpful for the reader to know whether "deficient" means null or hypomorphic.

6) It would seem from the structure of the crosses using ph(del) that all ph(del) mutant progeny would be males. Will the authors please comment on this with respect to their interpretations of the data and overall conclusions, especially with regard to comparisons of the two mutant backgrounds?

7) As the involvement of unknown second site mutations is always a concern, will authors please justify their use of samples that are homozygous or hemizygous across entire chromosomes.

8) Please indicate the numbers of trials and embryos used in their studies?

9) Regarding the statement, "In both mutants, Hox gene derepression started in a few cells, and the proportion of cells with derepression increased during later embryogenesis...", it seems that

the proportion of cells with derepression also decreases in some instances?

10) The following sentence is difficult to follow: "These results show that in PSs where every Hox gene of one complex is repressed, the first effect of Ph and Pc on Hox clusters folding can be detected before ectopic Hox gene transcription and affected whole Hox complexes, whereas the first effects on Hox genes derepression affected a minority of the Hox genes."

11) If possible, please change the colors used in the cartoons of embryos to demarcate the various PSs so that there is no confusion with meaning of the colors used in the various graphs. For example, in Figure 1, the colors of the bars in panels f-k seem at first to have been chosen to correlate with the PS segments.

12) The different fluorescent signals in the in situ images of Figure 1, Extended Data Figure 1, and Extended Data Figure 3 are difficult to see.

13) Figure 1: Panels f-k are mentioned in the text before panels a-e are mentioned.

14) Figures 2d and 2j: Is there a reason why the initiation of Ubx and AbdB expression is not noted?

15) Figures 2n-p and 3h-j: Please add explanations of the "triangles" to the legend.

16) Extended Data Figure 4: It would be useful to have a statistical statement on the differences or similarities among the four conditions.

17) Extended Data Figure 4: Please clarify the x axes in the figure by adding " μm ".

18) Extended Figure 7a-e needs quantitation.

19) Extended Figure 7a-e: Please indicate ages of the embryos.

20) There are a number of instances needing adjustments with respect to grammar. One example would be: "No effect on Hox *gene* transcription in these regions *was* revealed by..." Please scan the entire manuscript for other small grammatical issues.

Point-by-point response to reviewers comments

We wish to thank the reviewers for their positive evaluation of our work, their constructive suggestions and their detailed analysis. In response to their points, we have exhaustively revised the work, aiming at clarifying ambiguous points, improving the readability of this complex manuscript and adding new data and illustrations to convey the quality of the data and of the approaches, notably of DNA and RNA-FISH. This resulted in inclusion of new Figures S1 and S5-S7. Furthermore, old Figure S3 has been improved and expanded into two new Figures S3 and S4. Old Figure S7 has also been expanded with new data additions and split into two new Figures S11 and S12. As requested by one reviewer, old Figure S1 has been moved to the main manuscript as new Figure 1, but the imaging part (old S1 b-d) has been expanded and improved by the addition of 12 inset panels, illustrating examples of DNA-FISH data, in order to produce the new Figure S1. Furthermore, statistics has been improved, figure layout has been edited and the text has been carefully edited, following reviewer suggestions. The detailed changes are described below in response to individual reviewer points.

Reviewer #1

Genes silenced by Polycomb group (PcG) complexes are known to be more compact than actively transcribed genes. The PcG complex PRC1 drives this compaction. Is this compaction correlated with gene silencing? The authors present a series of experiments that show that, when PRC1 components are removed, chromatin becomes decompact prior to gene expression. The results show that loss of compaction itself does not necessarily lead to gene expression, but it does precede it, showing there is a correlation between compaction and gene silencing. The question addressed in this manuscript is an important one and the experimental approaches they used are appropriate. Below are a number of questions and suggestions to improve the overall quality of the manuscript.

We wish to thank the reviewer for showing appreciation for the question and the quality of the work. We address the specific points below.

1. *Fig. 1a-d - The FISH spots are almost invisible in the manuscript provided (even the source file). The poor quality of these images drives the reader to think “how were distances between the DNA-FISH spots measured?” The authors need to show zoomed insets of a few cells where two DNA-FISH spots are visible with a line indicating how the spot distances were measured. Also, it is difficult to determine where the parasegments are located. It is mentioned in the materials and methods that regions corresponding to the parasegments of interest were drawn manually. It would greatly help the reader to see these regions of interest outlined in a white dotted line. To better illustrate RNA FISH experiments, we zoomed on few regions to clearly show RNA FISH spots (Fig. 2 and Fig. S1). Some parasegments were outlined in Fig. S1, S3-S4 and Fig. 2. We also add 3 supplementary figures (Fig. S5-7) of DNA FISH images illustrating the main results of this manuscript.*
2. *Fig. 2 and 3- The labels “Pc xt109-control” and “Ph del-control” are confusing. Are the authors referring to the control genotype or the mutant minus the control genotype? It would be less confusing and more informative to show both mutant and control on the graphs. In the figure legends where the triangle plots are described as*

showing “the folding of BX-C” is confusing. A better wording would be “distance between BX-C genes.”

In order to avoid confusion, we specified in the legends that we are showing the difference between median distances observed in each of the mutant and its control. We also changed the labels “Pc xt109-control” and “Ph del-control” into “Pc xt109-cont Pcxt109” and “Ph del-cont Ph del” in order to clarify that for each mutant we took the appropriate control in order to calculate the distances. In addition, we show distance distributions separately for mutants and control embryos at the first (3h 50min – 4h 50min) and last (7h 20min - 12h) timepoints, but chose to maintain the plots showing the difference between mutant and control embryos in order to illustrate how the distance change evolves during development. In the figure legend of the triangle plots, we followed the reviewer suggestion and now added the sentence “plots of the three median distances, corrected for chromatic aberrations, between the promoters...”

3. *Fig. 4g - The genotypes and thus the colors, are in a different order than the previous six graphs. This confuses the reader, be consistent.*

We thank the reviewer for spotting the inconsistency. Fig. 4g of the previous ms is now Fig. 5 g-h. We now plotted the genotypes in the same order than the previous 6 graphs.

4. *Line 10 and line 41 - references are needed for the statement “ectopic transcription can open chromatin”*

The introduction section has been revised and this sentence, which is no longer needed for the logic of the text, has been removed.

5. *Line 71-75- authors show that ectopic Hox gene expression generally started earlier in Phdel embryos than in Pcxt109 embryos, but there is not any expression analysis of the Hox genes in Pcxt109 embryos for the early timepoint 3:50-4:50 hours after fertilization (Fig. 1 and Fig. Extended Data).*

We apologize if this was not clear but we studied this timepoint carefully and never observed Hox genes derepression in Pc xt109. We added ND in the figures and in the statistical table related to this experiment to avoid confusion.

6. *Line 127- authors claim that the nuclear Pc distribution became diffused in Phdel embryos whereas Ph still accumulates in foci in Pcxt109. This seems to be an overstatement from the image they provided (extended data 7 a-c). The number of Ph foci is drastically reduced from WT to Pcxt109. The authors need to show quantification of the Ph and Pc spots in WT vs mutants in several embryos.*

We understand this concern and therefore, in the revised ms, we have quantified the change in nuclear Pc distribution in *ph del* mutant embryos and the change of nuclear Ph distribution in *Pc Xt109* embryos (Fig. S11 d, e). These quantifications show that the distribution of Ph in *Pc* mutant embryos is not significantly different compared to controls, whereas Pc is significantly reduced in *ph* mutant embryos.

7. *Fig. S1 legend - Mention that a) the images shown are maximum projections of confocal images, b) FISH spots are measured in three dimensions and c) that the density of FISH spots is measured by area of the parasegment.*

We thank the reviewer for allowing us to improve clarity. Indeed, in the revised manuscript we mention that RNA FISH pictures are maximum intensity projections of confocal images. We also mention in the legend of Fig.1 that “the density of RNA FISH spots is calculated by dividing the number of spots by the area of the parasegment”. In fig. S5-7, we state that the distance between loci were measured in three dimensions between the centre of mass of each FISH spot.

8. *Fig. S3 - Conclusions cannot be drawn from the poor quality images provided.*
In the revised ms, we show four embryos at higher magnification (Fig. S3-4) and, for clarity, we outlined the position of specific PSs by dashed lines.

9. *Supp. Methods RNA FISH - List the fluorophores used for labeling probes. Please provide the final concentration of probe used for the hybridization in pmol fluor/μl.*
Done as requested.

10. *Supp. Methods Immuno-DNA FISH - What are the dilutions used for each antibody? What company produced the secondary antibodies used in the experiments?*
In the methods section, we now added the concentrations of antibodies and the source of secondary antibodies.

11. *Supp. Methods microscopy and image analysis – Were the control and mutant genotypes were imaged and analyzed under the same parameters? This is critical to the integrity of the conclusions regarding FISH spot density, as different densities could be obtained by using different imaging parameters for control and mutant samples.*

This is a very important comment. Controls and mutants were indeed acquired with the same setting and this is now indicated in the methods section, as requested.

Reviewer #2

This work is interesting, potentially unravelling the link between PRC1-mediated chromatin architecture and transcriptional activation. I have some concerns about the extent to which the current molecular data support the final conclusion. In fact, the main conclusion of this work depends on a comparison between RNA-FISH and DNA-FISH, experimentally performed in parallel. While it is reasonable to compare the same RNA-FISH analysis on different strains, the conclusions about chromatin structure and transcription can not be drawn only on the basis of DNA-FISH/RNA-FISH comparison because the two technologies could have different resolutions. Moreover, DNA-FISH data was presented as an average of the minimal distances between two spots in the parasegment-specific population and did not provide an estimation at the single cell level of the percentage of nuclei lacking DNA interaction (images were not presented!). A combo DNA/RNA FISH could prove that in the PcG mutants, nuclei presenting different chromatin conformations do not show local

transcription. Considering that the work did not provide strong evidence for its conclusions I cannot support a publication in Nature Communication.

We thank the reviewer for finding the work of interest. In response to the main criticism, let us present the reasoning that lead us to trust our conclusions. First, both RNA FISH and DNA FISH experiments are, of course, single-cell methods. Hox genes in flies express in a highly deterministic manner. In each PS corresponding to the domain of expression of each Hox gene, the overwhelming majority of the cells expressed the corresponding Hox gene (Fig. S1). To quantify transcription, we used the density of RNA FISH spots which, as defined in the legend of Figure 1, directly depends on the number of cells in which transcript signals are detected. To monitor the folding of HOX clusters, we performed DNA FISH experiments recognizing 3 loci per cluster and then we measured the distances between FISH spots corresponding to each locus. Every measurement is done in single cell and, in addition to average values, we present box plots to show entire distributions of distances measured in cells of given parasegments. In order to comply with the request from the reviewer, in the revised version of the ms we present extensive examples of single-cell data that allow appreciating the imaging quality. Let us also comment on the lack of combined DNA and RNA FISH. While this would of course be a interesting approach on paper, two reasons suggested us not to take this path. First and most important: our critical conclusions concern embryonic regions and developmental times in which no expression is detected for Hox genes, neither in wild type, nor in mutants. Therefore, a combo would not add power to the conclusion that chromatin opening precedes transcriptional derepression, which is the main conclusion of the work. Second, the technical difficulties of implementing a combined DNA & RNA FISH on whole mount embryos make this strategy less suited in our particular case. Indeed, a combination of the two methods requires a sequential approach in which one performs RNA-FISH, then makes image acquisition of the data, and then proceeds to DNA-FISH. During this complicated two-step procedure including a long image acquisition step prior to DNA-FISH, some embryos might move or be damaged. This will ultimately reduce the number of usable data and thus the statistics. However, the very large amount of measured distances, i.e. an extremely strong statistics, is actually a very strong aspect of our present analysis, which allows us to safely draw conclusions for distance distributions even when differences are not very large. Reducing the statistics would prevent us from doing this and thus the loss in statistical power might undercut the potential gain coming from the fact of disposing of data in the same cell.

- 1. In RNA-FISH, absence of the spot does not prove the absence of transcription, thus authors could underestimate a subpopulation of cells with lower transcription in PcG mutants. Parasegment microdissections followed by single cell RNA analysis could untangle this point.*

We agree that absence of spot signals does not prove an absence of transcription. However, many former studies have analyzed ectopic Hox gene expression in PRC1 mutant embryos. Our data confirm the previous results of Hox gene deregulation, but we actually detect s earlier than previous studies. Moreover, we also detected an earlier expression of two other genes, *dac* and *vg*, compared to previous reports. Together, these data suggest that our sensitivity is the highest achieved to date. Noteworthy, previous studies have also analysed the expression of Hox genes, *dac* and *vg* at high throughput in whole embryos and classified in particular *dac* and *vg* as very lowly expressed at the developmental time points in which we can detect them by

RNA-FISH. This suggests again that we can detect very low amount of transcripts by RNA-FISH. It is still possible that in some of the cells here and there we missed to detect ectopic transcription, but we think that these sporadic events could not explain the general Hox chromatin opening that we observe over whole spatial embryonic domains before any transcriptional detection.

2. *I cannot find in the figure legend or in the methods the number of nuclei that were taken into consideration in the RNA-FISH and DNA-FISH analyses. Only the number of embryos is indicated.*

We thank the reviewer for this comment. The detailed numbers of measurements for each of the experiments are now shown in the statistical information section.

3. *Extended Fig 3 h and j: I am not sure that the green signal is, as expected, inside nuclei.*

The green signal present outside PSs (Fig. S4) correspond to the yolk of the embryo, a staining artefact that is generally observed in fly RNA FISH. In the revised ms, we present pictures of RNA FISH at higher magnification to better show signals inside cell nuclei.

4. *Extended Fig 3: the image “j” is cut.*

We thank the reviewer for spotting this apparent issue which allows us to clarify the situation for non-fly experts. Of course, no image splicing has been applied throughout the work. Instead, the embryo is too big to fit single microscope frames. Therefore, we have to combine 3 (or more) stacks to reconstitute a full embryo. For transparency, in the revised ms, dashed-lines indicate where stacks were merged.

5. *Images showing DNA-FISH were showed only in extended data Figure 7 with only one probe. Why? The authors should document their experiments and quantification with representative images.*

We totally agree with the reviewer and, in the revised ms, we present 3 extra supplementary figures (Fig. S5-7) showing panels with 3 colours DNA FISH experiments by way of examples of the images that were used to measured 3D distances.

6. *Figure 4g: Authors decided to normalize the Pc signal with “the average intensity inside the cell nuclei”. However, they described a diffuse Pc signal in Ph mutant in the extended data Figure 7b. Thus the normalization inside nucleus could have a bias.*

We now show a quantification of the average intensity measured inside cell nuclei in Fig. S11f. The results show that the difference between control and mutants is not significant. Indeed, part of the Pc signal is nucleoplasmic even in the WT.

7. *The Pc diffusion showed in the extended data Figure 7b is not present in the same strain in extended data Figure 7d.*

The image shown in (old) extended data Figure 7b has been obtained after immunostaining experiment, whereas the image shown in extended data Figure 7d presented an immuno-FISH experiment which usually exhibit a lower background inside cell nuclei, due to harsher permeabilization high temperature treatment of the nuclei that are required for hybridization. These background differences are inherent to the methods, such that immuno-FISH is not suitable to study the diffusible nucleoplasmic component of proteins. Instead, the more stably chromatin associated proteins survive immuno-FISH allowing to measure differences in association of PcG components to specific genes in immuno-FISH.

8. *Super resolution analysis could improve the quality of immuno-FISH experiments.*

Super resolution microscopy would improve distance measurement if we measured distance between FISH spots with the same fluorochrome. However, we used 3 different fluorochromes and we only measured distance between the centroids of 3 spots corresponding to each fluorochrome. In this case, high quality conventional confocal microscopy already provides centroid determination precision with a resolution well below the Abbe limit of resolution (few tens of nanometers).

9. *As control authors should measure the localization of Pc in FISH analysis of HOX regions that does not change conformation, such as AbdA-AbdB.*

This was an excellent suggestion. In the revised ms, we show the enrichment of Pc and Ph at Hox gene where they are expressed (Fig. 5g-h; PS2 for *Scr* and PS9-12 for *adbA*). As expected, we found a much smaller changes in this case.

10. *An alternative experiment that can measure the amount of Pc protein in the close proximity of DNA of interest is a variant of the Proximity Ligation Assay (PLA) (Gustafsdottir, SM et al., PNAS 2007), used to quantify Protein/DNA binding in immunofluorescence.*

PLA would be of interest to try and quantify differences in the amount of Pc or Ph binding to the target sites. In our case however, we are specifically asking whether the loci under study form higher-order architectures. A sizeable amount of the protein in PcG foci associated to the gene of interest could be located at considerable distances (in the hundred nm range) from the binding site, distances that would not be detected in PLA assays. These higher-order foci might nevertheless play a role in stabilizing silencing by forming nuclear compartments of high PcG concentration. For this reason, we carefully quantified the distance of the DNA probes from the closest Pc or Ph site (Figure 1) or the intensity of PcG protein signals (Figures 5 and S11) in the experiments involving the analysis of PcG protein amounts.

Reviewer #3

In this study, the authors undertook a massive exercise to image Hox cluster loci in Drosophila through early stages of embryogenesis. They compared the effect of deleting two different components of canonical PRC1 – Ph and Pc – on intergene-distance between Hox cluster genes. The main outcome is that the authors were nicely able to separate two known functions of PRC1 – chromatin compaction and gene repression, in time. They show that decompaction of Hox cluster genes precedes transcriptional activation. Further they confirmed the conclusion previously made by them and by other groups that PRC1 loss leads to decompaction of the large Hox gene clusters. This is an important addition to the literature concerning the mechanisms that repress expression to control developmental progression.

We thank the reviewer for this positive evaluation and address the specific points raised by this reviewer below.

1. *This is an elaborately designed study with several time points and a lot of data. The data, however, are presented in a manner that is hard to absorb. Thus, it takes the reader a significant amount of effort to arrive at the conclusions that the authors make. One issue is that the expression data (most of Fig. 1) is presented in three dimensional histograms, while the compaction data (Figs. 2 and 3) is presented in box plots and graphs. The disparate means of presentation makes*

direct comparison hard to do. Perhaps the authors might take especially central examples and do a plot of expression vs compaction side by side in the same format.

We appreciate this comment and did our best to improve the readability and the clarity of the message and the data in the revised manuscript. The reason for presenting the expression data in a different format from the 3D distances is that these are different data types. In the case of expression, we quantify numbers of nuclei in which we detect nascent transcripts in various conditions. In each nucleus, we have a binary “yes”/“no” result. In the case of distances, we measure each of them and can analyse a continuous distance distribution. However, to help the reader compare expression time with 3D distance changes, we added arrows in Fig. 3d-f, 3 j-l and 4 d-f which indicate the timing of the earliest ectopic Hox gene transcription. On these graphs, we can directly compare the timing of ectopic Hox genes expression and the timing of structural effects.

- 2. The actual microscopy images in Figure 1 are nearly impossible to appreciate and one has to rely entirely on the quantification. The entire study relies heavily on measurements of intergene distance between Hox genes. It is important to show the reader examples of those measurements, and what the primary data looks like when comparing a compacted and a more open setting.*

In the revised ms, we zoomed on few regions to clearly show RNA FISH spots (Fig. 2 and Fig. S1). We also present 3 supplementary figures (Fig. S5-7) showing examples of 3 colours DNA FISH experiments used to measured 3D distances.

- 3. The title suggests that compaction has a causal role in preventing ectopic transcription. While the authors have convincingly demonstrated that Hox decompaction precedes Hox transcription, I do not think that we have learned from this study that compaction causes gene silencing or prevents gene expression. As the author’s themselves find and later suggest, the decompacted Hox genes are not immediately expressed, probably due to the lack of an appropriate activator. Perhaps expand on this topic and offer alternative explanations.*

We agree with the reviewer comment and, according to this, we have revised the title to avoid making causal claims that are not totally demonstrated. Furthermore, in the discussion section we have elaborated about the possible requirement of early compaction of Hox clusters for later maintenance of Hox genes silencing.

- 4. This is not necessary for this study, and technically a very tough thing to do, but it would be nice and very interesting to see if the general principles described in this study are unique to the uniquely organized Hox genes or whether they are also true for other, more typically organized, PRC1 target loci.*

As the reviewer states, this is a very important and complicated work which is actually the core of our future project. Hox genes are the largest PcG target chromatin domains and we expect that the study of different categories of targets with progressively smaller size and number of PcG binding sites should help understand whether locus compaction is a general feature of Polycomb-mediated gene silencing

Reviewer #4

"PRC1-dependent compaction of Hox gene clusters prevents transcriptional derepression during early Drosophila embryogenesis" by Thierry Cheutin and Giacomo Cavalli is an impressively detailed study that is even more remarkable given the small number of authors (two). It is focused on a very specific question, which is whether the repressive effect of PRC1 on gene activity is the result of chromatin compaction. To address this question, the authors examine the temporal relationship between chromatin compaction and gene silencing/activation at the bithorax and Antennapedia complexes (BX-C and ANT-C) in wild-type embryos as well as in embryos mutant for either polyhomeotic (ph) or Polycomb (Pc). The authors report a correlation between chromatin compaction and gene silencing and, furthermore, that the null state of ph or Pc leads to chromatin decompaction prior to the activation of Hox gene expression from the BX-C and ANT-C. These observations lead the authors to conclude that the mechanism by which PRC1 represses gene expression is through the compaction of chromatin. The figures show meticulous data from countless DNA and RNA fluorescent in situ hybridization (FISH) assays targeting regions within the BX-C and ANT-C and, overall, the narrative is well-written. Even so, there are a number of issues that deserve attention from the authors. These are noted below:

We thank the reviewer for the interest in our work and the valuable inputs. We reply to each specific comment below:

1. *The most pressing technical issue is whether the method used to assess transcriptional activity is sensitive enough to justify the strength of the conclusions. Thus, the authors are encouraged to quantify, and/or consider in detail, the sensitivity of their assays and then adjust the strength of their conclusions accordingly. For example, how certain are the authors that their assays could have detected a very low level of transcription or perhaps even paused polymerases?*

This is a relevant point which was also raised by reviewer #2. For a detailed answer, see #2, point1. In short, we certainly can not exclude paused polymerase or rare transcription events, but we do have strong argument to say that we detect transcription in a sensitive manner. Nevertheless, according to the reviewer suggestion, we did adjust the phrasing to avoid making excessive claims throughout the manuscript.

2. *Similarly, the underlying logic of the study - that decompaction prior to gene activation in a mutant background demonstrates that chromatin compaction is the mechanism for silencing - is not as strong as the authors argue it to be. For example, formally speaking, PRC1 might independently induce to two outcomes, one being chromatin compaction and the other being gene silencing...*

We agree with the suggestion to avoid overstating conclusions. We thus changed the title of the ms and several statements throughout the text in order to provide a more balanced introduction, discussion and conclusion from our work.

3. *There are two instances where expression in ph(del) embryos seems to have been detected at the earliest time point (3:50 - 4:50 hr AEL): Ubx in PS2-4 and Antp in the head. As these observations are so directly relevant to the conclusions, have*

the authors looked at an earlier time point? If not, the authors may wish to consider a more serious discussion of these exceptions.

We analyzed earlier timepoints and we did not see ectopic expression of neither *Ubx* or *Antp*. However, we prefer not to stress this point since at earlier time points there is some residual maternal component (most likely depending on the Ph-d subunit) and therefore the lack of effect might reflect the presence of sufficient repressor on the chromatin.

4. *Will the authors please comment on the potential relevance of maternal effects to the interpretation of their data and their conclusions? In addition, might maternal effects explain the more obvious effects of the mutant backgrounds later in embryogenesis?*

In the revised ms, we now commented on the potential relevance of maternal effects. We do not think that maternal effects explain most of our data since, at the earliest data point we analyse, Pc and Ph are already undetectable in mutant embryos.

5. *Please provide a better description of *ph(del)* and *PC(XT109)*, as these are critically important mutations. In particular, it would be helpful for the reader to know whether "deficient" means null or hypomorphic.*

In the result section, we now mention that Ph del and Pc xt109 are null mutants.

6. *It would seem from the structure of the crosses using *ph(del)* that all *ph(del)* mutant progeny would be males. Will the authors please comment on this with respect to their interpretations of the data and overall conclusions, especially with regard to comparisons of the two mutant backgrounds? ... continues with 7...*
7. *As the involvement of unknown second site mutations is always a concern, will authors please justify their use of samples that are homozygous or hemizygous across entire chromosomes.*

These are very relevant points and we therefore carefully analysed multiple control embryos. Specifically, we analyzed the expression of the eight Hox genes in 3 control lines (A "wild type" line Wi, a control of Phdel, a line containing the FKG balancer chromosome, and a control of Pc xt109, which contains a TKG balancer chromosome). We did not detect any difference in Hox genes expression among the three controls. Similarly, we did not observe any difference in the folding of Hox clusters between these 3 *Drosophila* lines. We now mentioned this point in the methods section of the revised ms.

8. *Please indicate the numbers of trials and embryos used in their studies?*

Numbers of measurements are now shown in the statistical information section.

9. *Regarding the statement, "In both mutants, Hox gene derepression started in a few cells, and the proportion of cells with derepression increased during later embryogenesis...", it seems that the proportion of cells with derepression also decreases in some instances?*

The proportion of cells with derepression only decreases in late embryogenesis. Hox proteins can regulate Hox gene transcription since a posterior Hox gene

usually silences anterior ones (a phenomenon called Hox posterior prevalence or posterior dominance and reflecting cross-regulation of Hox gene expression). Therefore, this late effect probably relies on Hox transcriptional cross-regulation.

10. *The following sentence is difficult to follow: "These results show that in PSs where every Hox gene of one complex is repressed, the first effect of Ph and Pc on Hox clusters folding can be detected before ectopic Hox gene transcription and affected whole Hox complexes, whereas the first effects on Hox genes derepression affected a minority of the Hox genes."*

We corrected this sentence in the revised MS.

11. *If possible, please change the colors used in the cartoons of embryos to demarcate the various PSs so that there is no confusion with meaning of the colors used in the various graphs. For example, in Figure 1, the colors of the bars in panels f-k seem at first to have been chosen to correlate with the PS segments.*

We now changed the colors used in the cartoons of embryos to demarcate the various PSs in the revised ms, according to the reviewer suggestion.

12. *The different fluorescent signals in the in situ images of Figure 1, Extended Data Figure 1, and Extended Data Figure 3 are difficult to see.*

We thank the reviewer for this remark. In the revised ms, we now show 4 embryos at higher magnification (Fig. S3-4) and we zoomed in on few regions to show RNA FISH spots more clearly (Fig. 2 and Fig. S1).

13. *Figure 1: Panels f-k are mentioned in the text before panels a-e are mentioned.*

This has been duly corrected in the revised version.

14. *Figures 2d and 2j: Is there a reason why the initiation of Ubx and AbdB expression is not noted?*

As requested, we now added arrows indicating the initiation of *Ubx*, *abdbA* and *AbdB* in Fig. 3d and j.

15. *Figures 2n-p and 3h-j: Please add explanations of the "triangles" to the legend.*

Done, as requested.

16. *Extended Data Figure 4: It would be useful to have a statistical statement on the differences or similarities among the four conditions.*

As requested, we added statistical statements in new Fig. S8 (old Fig S4).

17. *Extended Data Figure 4: Please clarify the x axes in the figure by adding " μm ".*

Done as suggested.

18. *Extended Figure 7a-e needs quantitation.*

In the revised ms, we have quantified the change in nuclear Pc distribution in Ph del and the change of nuclear Ph distribution in Pc xt109 as requested (Data are presented in the new Fig. S11 d-f).

19. Extended Figure 7a-e: Please indicate ages of the embryos.

We thank the reviewer for spotting this and have indicated the age of the embryos in the legend of the figure (new Fig. S11)

20. There are a number of instances needing adjustments with respect to grammar.

One example would be: "No effect on Hox *gene* transcription in these regions *was* revealed by..." Please scan the entire manuscript for other small grammatical issues.

We went through the English and corrected mistakes.

Reviewers' Comments:

Reviewer #2:

In the revised version of the manuscript "Loss of PRC1 induces higher-order opening of Hox loci independently of transcription during early Drosophila embryogenesis" my major complaints were not addressed.

Although I know that DNA/RNA FISH is a complicated experiment, excellent groups were able to publish the combination of the two technologies (Curr Protoc Hum Genet. 2013 "A multifaceted FISH approach to study endogenous RNAs and DNAs in native nuclear and cell structures" Byron M1, Hall LL, Lawrence JB, doi: 10.1002/0471142905.hg0415s76; Methods Mol Biol. 2008 "Combined immunofluorescence, RNA fluorescent in situ hybridization, and DNA fluorescent in situ hybridization to study chromatin changes, transcriptional activity, nuclear organization, and X-chromosome inactivation"

Chaumeil J1, Augui S, Chow JC, Heard E, doi: 10.1007/978-1-59745-406-3_18; Methods Mol Biol. 2010 "Detection of nascent RNA transcripts by fluorescence in situ hybridization" Brown JM1, Buckle VJ, doi: 10.1007/978-1-60761-789-1_3).

The second issue was about the quantification and interpretation of immuno-FISH experiments, since the quality of the immuno-FISH images is low. I suggested to perform super-resolution for the immunoFISH experiment (not for FISH experiments) to better quantify distances of FISH spot from closest PcG body. PLA (protein-DNA) experiments could have been an alternative approach to quantify these distances.

Reviewer #3:

This revision addresses the comments I made on the initial submission in a satisfactory manner. The strength of this paper remains that it addresses a key question in repression by the Polycomb-Group: how is chromatin compaction related to transcription? Is compaction/decompaction independent of transcription, consistent with it being causal, or is it an outcome of transcription. The strength of the paper is that it makes a compelling argument that decompaction occurs prior to transcription.

The key issue is whether the sensitivity of the transcription measurements is sufficient to make a robust argument, a point clearly made by reviewers 2 and 4, and what I was trying to get at in my poorly worded comment 1 of my initial review. I found the arguments made on this point in the rebuttal to be good. It is clear that a completely rigorous statement cannot be made at this point with current technology, but the paper moves the ball down the field by making a strong argument that the major conclusion is very likely correct. I thought the prose in the first paragraph of the discussion did a good job of framing the argument.

In sum, this paper addresses a fundamental point in developmental regulation, does a thorough job of it, and reaches an important conclusion. It is of sufficient general interest, and presents an important data set. I think it should be published.

Reviewer #4:

None

Point-by-point response to reviewers comments

Reviewer #2 (Remarks to the Author):

In the revised version of the manuscript “Loss of PRC1 induces higher-order opening of Hox loci independently of transcription during early *Drosophila* embryogenesis” my major complains were not addressed.

Although I know that DNA/RNA FISH is a complicated experiment, excellent groups were able to publish the combination of the two technologies (Curr Protoc Hum Genet. 2013 “A multifaceted FISH approach to study endogenous RNAs and DNAs in native nuclear and cell structures” Byron M1, Hall LL, Lawrence JB, doi: 10.1002/0471142905.hg0415s76; Methods Mol Biol. 2008 “Combined immunofluorescence, RNA fluorescent in situ hybridization, and DNA fluorescent in situ hybridization to study chromatin changes, transcriptional activity, nuclear organization, and X-chromosome inactivation” Chaumeil J1, Augui S, Chow JC, Heard E, doi: 10.1007/978-1-59745-406-3_18; Methods Mol Biol. 2010 “Detection of nascent RNA transcripts by fluorescence in situ hybridization” Brown JM1, Buckle VJ, doi: 10.1007/978-1-60761-789-1_3).

The second issue was about the quantification and interpretation of immuno-FISH experiments, since the quality of the immuno-FISH images is low. I suggested to perform super-resolution for the immunoFISH experiment (not for FISH experiments) to better quantify distances of FISH spot from closest PcG body. PLA (protein-DNA) experiments could have been an alternative approach to quantify these distances.

To address the criticisms of the reviewer concerning technical limitations of our manuscript, we have commented on these limitations in the discussion of the revised manuscript, where we discuss that a combined DNA and RNA FISH would directly allow to compare chromatin folding between cells expressing one Hox gene and the ones that do not. This approach would help to better characterize the late effect of Ph and Pc on chromatin folding of Hox cluster, where Hox gene ectopic expression occur in a sub-population of cells of one parasegment. However, our critical conclusions that chromatin opening precedes transcriptional derepression concerns early developmental times and embryonic regions in which no expression is detected for Hox genes, neither in wild type, nor in mutants. Therefore, we suggest that combined DNA/RNA FISH would not help reaching our critical conclusion.

The immuno-FISH experiments have been done mainly to discriminate Ph and Pc mutant embryos from control ones. We only measured distances of FISH spot from the closest Pc/Ph foci (Fig. 1h and i) along the A/P axis to perform a relative comparison of Hox gene position within Pc/Ph foci. As mentioned in the manuscript, this result mainly confirms previously published studies. Performing super-resolution microscopy would greatly improve the description of the distribution of Pc and Ph in these nuclear foci. We have introduced a statement in the ms to discuss this point and applying this approach could be an interesting future project. However, this

is beyond the scope of this work since we mainly focus on chromatin folding of Hox clusters to demonstrate that cPRC1 compaction occurs before ectopic transcription.

Reviewer #3 (Remarks to the Author):

This revision addresses the comments I made on the initial submission in a satisfactory manner. The strength of this paper remains that it addresses a key question in repression by the Polycomb-Group: how is chromatin compaction related to transcription? Is compaction/decompaction independent of transcription, consistent with it being causal, or is it an outcome of transcription. The strength of the paper is that it makes a compelling argument that decompaction occurs prior to transcription.

The key issue is whether the sensitivity of the transcription measurements is sufficient to make a robust argument, a point clearly made by reviewers 2 and 4, and what I was trying to get at in my poorly worded comment 1 of my initial review. I found the arguments made on this point in the rebuttal to be good. It is clear that a completely rigorous statement cannot be made at this point with current technology, but the paper moves the ball down the field by making a strong argument that the major conclusion is very likely correct. I thought the prose in the first paragraph of the discussion did a good job of framing the argument.

In sum, this paper addresses a fundamental point in developmental regulation, does a thorough job of it, and reaches an important conclusion. It is of sufficient general interest, and presents an important data set. I think it should be published.

We wish to thank the reviewer for this positive evaluation.